# Unified AI framework to uncover deep interrelationships between gene expression and Alzheimer's disease neuropathologies

Nicasia Beebe-Wang [1], Safiye Celik[2], Ethan Weinberger[1], Pascal Sturmfels[1], Philip L. De Jager [3], Sara Mostafavi[1,4,5 ✉] & Su-In Lee [1,5 ✉]

Deep neural networks (DNNs) capture complex relationships among variables, however, because they require copious samples, their potential has yet to be fully tapped for understanding relationships between gene expression and human phenotypes. Here we introduce an analysis framework, namely MD-AD (Multi-task Deep learning for Alzheimer's Disease neuropathology), which leverages an unexpected synergy between DNNs and multi-cohort settings. In these settings, true joint analysis can be stymied using conventional statistical methods, which require "harmonized" phenotypes and tend to capture cohort-level variations, obscuring subtler true disease signals. Instead, MD-AD incorporates related phenotypes sparsely measured across cohorts, and learns interactions between genes and phenotypes not discovered using linear models, identifying subtler signals than cohort-level variations which can be uniquely recapitulated in animal models and across tissues. We show that MD-AD exploits sex-specific relationships between microglial immune response and neuropathology, providing a nuanced context for the association between inflammatory genes and Alzheimer's Disease.

[1] Paul G. Allen School of Computer Science and Engineering, University of Washington, Seattle, WA, USA. [2] Recursion Pharmaceuticals, Salt Lake City, UT, USA. [3] Department of Neurology, Center for Translational and Computational Neuroimmunology, Columbia University Medical Center, New York, NY, USA. [4] Department of Statistics, University of British Columbia, Vancouver, BC, Canada. [5] These authors jointly supervised this work: Sara Mostafavi & Su-In Lee. ✉email: saramos@cs.washington.edu; suinlee@cs.washington.edu

Alzheimer's disease (AD), the sixth leading cause of death in the United States, is a degenerative brain condition with no known treatment to prevent, cure, or delay its progression. Primary challenges to treating and preventing AD include extensive heterogeneity in the clinicopathologic state of older individuals[1] and limited knowledge about genetic and molecular drivers and suppressors of AD-related (amyloid and tau) proteinopathies and AD dementia[2]. Recent efforts to identify molecular mechanisms underlying AD and its progression focus on two complementary approaches. First, the assembly of large genome-wide association studies ($N > 100$ K subjects) enabled case/control analyses of genetic variants correlated with a clinical diagnosis of AD. Interestingly, some identified variants have implicated tau protein binding, amyloid precursor protein (APP) metabolism, or immune pathways that play a role in their aggregation and/or uptake[3–5]. These results reinforce the need for detailed investigations of the drivers of neuropathological variation across individuals. Second, moderate-scale postmortem transcriptomic studies have investigated molecular correlates of a richer set of phenotypic and neuropathological outcomes[6–9]. Early work in this domain examined pairwise correlations among gene expression levels and AD-related traits[10] or a diagnosis of AD[11]. More recent attempts have focused on learning statistical dependencies among gene expression using AD expression data collected from one cohort in order to infer gene regulatory networks[7] or co-expressed modules[6] associated with AD-related phenotypes (see Methods for details). The relative scarcity of brain gene expression data collected from each cohort has posed a challenge to the use of complex models, such as deep neural networks.

The collection of postmortem brain RNA-sequencing datasets, assembled by the AMP-AD (Accelerating Medicines Partnership Alzheimer's Disease) consortium, provides a unique opportunity to combine multiple datasets in an integrative analysis. Previous work has applied existing co-expression methods to each dataset and used consensus methods to identify consistent gene expression modules across datasets[9]. To our knowledge, there has not yet been a unified approach to learn a single joint model that incorporates multiple AMP-AD datasets, which would enable the use of all samples to capture intricate interactions between gene expression levels and neuropathological phenotypes. A unified approach has been hindered by: (1) the need for "harmonized" phenotypes consistently measured across datasets, and (2) the limitation of current analysis methods that focus on linear relationships between variables (e.g., module analysis[9]), which capture only broad patterns in gene expression data. These often correspond to cohort-level variations which consequently obscure true disease signals[12]. To circumvent this issue, one approach has been to identify modules separately across brain regions and cohorts before performing using a consensus approach to cluster them[9].

Here, we develop MD-AD (Multi-task Deep learning for Alzheimer's Disease neuropathology), a unified framework for analyzing heterogeneous AD datasets to improve our understanding of an expression basis for AD neuropathology (Fig. 1a–d). Unlike previous approaches, MD-AD learns a single neural network by jointly modeling multiple neuropathological measures of AD (Fig. 1a), and hence it incorporates the largest collection of postmortem brain RNA-sequencing datasets assembled to date. The combined AMP-AD dataset contains 1758 samples distributed across nine brain regions, which are labeled with up to six neuropathological outcomes that are sparsely available across cohorts (Fig. 1e). This unified framework has key advantages over separately trained models. First, MD-AD can accommodate sparsely labeled data, which is a natural characteristic of datasets aggregated through consortium efforts (Fig. 1e). Even if different phenotypes only partially overlap in the measured samples, each sample contributes to the training of both phenotype-specific and shared layers (Fig. 1a). Predicting multiple outcome variables at once biases shared network layers to capture relevant features of all those outcome variables (here, neuropathological phenotypes) at the same time[13]. This is of critical importance in our application: each neuropathological phenotype represents a different noisy measurement of the same underlying true biological process, and, as we demonstrate, joint training with these phenotypes allows MD-AD to average out the noise to extract the true hidden signal. In addition, the increased sample size from combining cohorts (in our case, doubling the number of samples available from any individual study) facilitates using deep learning models, which are expressive and able to capture complex non-linear interactions among features. By composing layers of functions, deep neural networks collapse correlation patterns present in input data at intermediate layers in a way that is useful for prediction[14]. In particular, multi-layer perceptrons (MLPs) have been used to effectively perform disease classification and prediction from gene expression data[15–17]. However, training separate MLPs for each neuropathological phenotype (Supplementary Figure 1a) has limited scope: it can utilize only the samples measured for a specific phenotype, and it cannot share information across related phenotypes. We demonstrate that MD-AD's joint training approach improves prediction accuracy, enabling its predictions to generalize across species and tissue types (Fig. 1b).

An obvious drawback of deep neural networks is their black-box nature, making it difficult to biologically interpret gene-phenotype associations that have been learned by a model. We present two ways to address this challenge. First, MD-AD adopts a well-known feature attribution method[18], which quantifies how much each input variable (here, gene expression level) contributes to a prediction (here, a neuropathological phenotype) to identify genes and pathways relevant to each neuropathological phenotype (Fig. 1d). Second, because MD-AD is a deep learning model, we can interpret its intermediate layers as biologically relevant high-level feature representation of gene expression levels and its predictions as the amalgamation of AD-specific molecular markers. The last shared layer of MD-AD can be viewed as a supervised embedding influenced by each neuropathological phenotype used during training. Thus, by interpreting this layer's embedding, we gain an understanding of model components and high-level dependencies between expression and neuropathology (Fig. 1c). We identify globally important genes not previously implicated in linear methods and then perform sex-specific analyses to explore implicitly captured non-linear effects among genes and their differing relationship with AD severity predictions.

In sum, the MD-AD framework makes the following contributions: (1) It is able to effectively impute accurate AD neuropathological phenotype predictions from broad compendia of heterogeneous brain gene expression data; (2) it produces learned representations that are more robust than separately learned models, improving generalizability to other datasets, species, and even tissue types; (3) it provides an improved understanding of interrelationships among molecular drivers of AD neuropathology that is missed by linear methods; and (4) from a biological standpoint, MD-AD highlights a sex-specific relationship between microglial immune activation and neuropathology.

## Results

**MD-AD provides a unified framework to learn a single model of multiple neuropathological phenotypes across multiple cohort datasets.** The MD-AD model takes as input brain gene

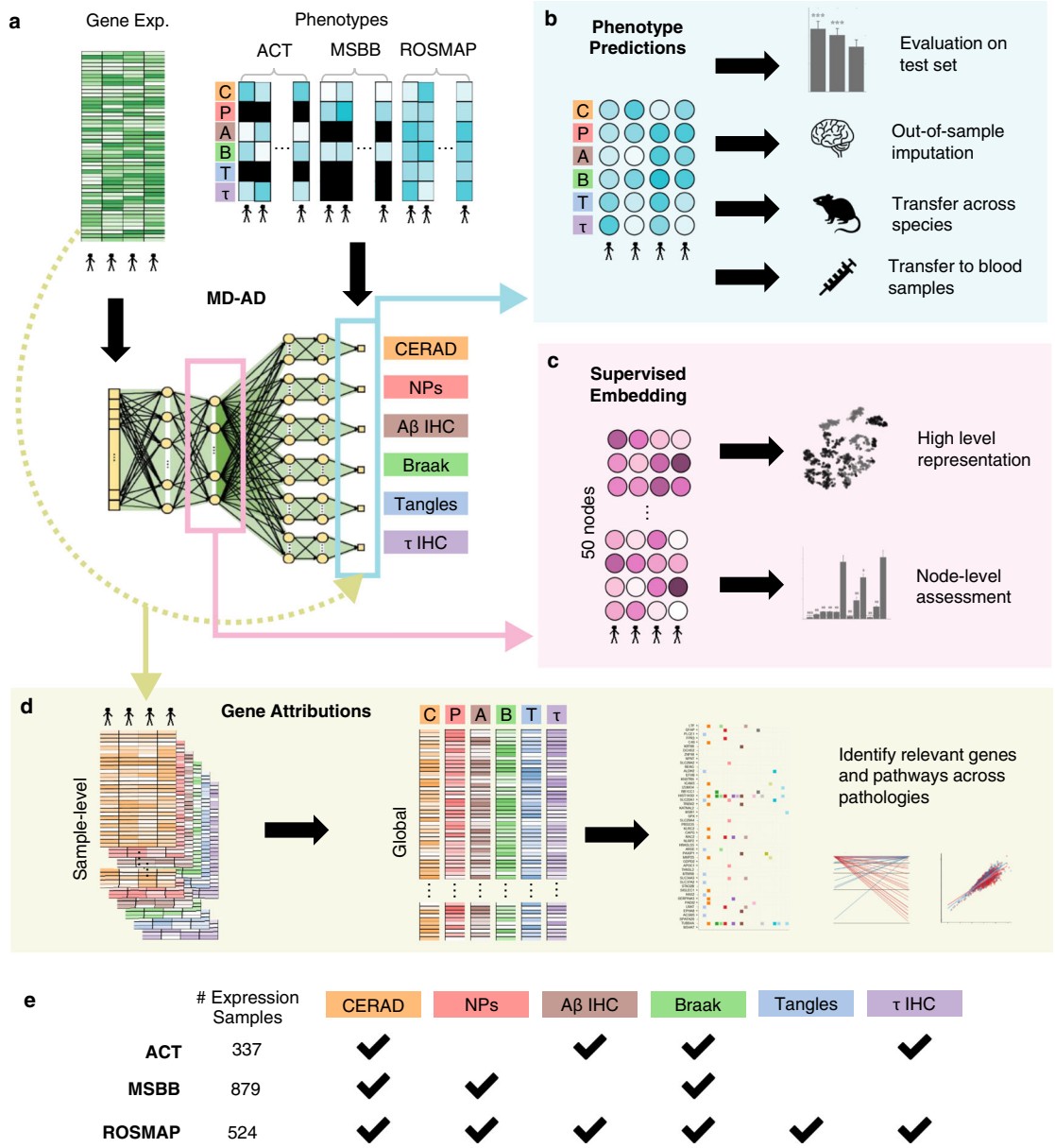

**Fig. 1 Overview of the MD-AD (Multi-task Deep learning for Alzheimer's Disease neuropathology) method and analyses. a** Overview of the MD-AD framework: MD-AD is trained to predict six neuropathology phenotypes simultaneously from brain gene expression samples. During model training, samples do not need to have all available phenotypes; they influence only the layers for which they have labels (including shared layers). **b** Illustrates out-of-sample datasets we used to validate MD-AD's predictions. **c** Illustrates analyses used to validate the last shared layer of MD-AD. **d** By using model interpretability methods, we highlight genes relevant to MD-AD's predictions. Further analyses reveal non-linear effects among genes and their relationship with AD severity prediction. **e** Overview of data available from each cohort.

expression profiles and simultaneously predicts several AD-related neuropathological phenotypes (Fig. 1a). In particular, the model is trained on expression data from the Religious Orders Study/Memory and Aging Project (ROSMAP)[6,19,20], Adult Changes in Thought (ACT)[21], and Mount Sinai Brain Bank (MSBB)[22] cohort studies, which together have 1758 gene expression profiles for 925 distinct individuals (with no participant overlap between cohorts). These data are normalized for study batch (Methods, Supplementary Figure 1d)[23]. As shown in Fig. 1a, the MD-AD model simultaneously predicts six AD-related neuropathological phenotypes: three related to amyloid plaques and three to tau tangles. The former include (1) Aβ immunohistochemistry (IHC): amyloid-β protein density via immunohistochemistry, (2) NPs: neuritic amyloid plaque counts

from stained slides, and (3) CERAD score: a semi-quantitative measure of neuritic plaque severity[24]. The latter include (4) τ IHC: abnormally phosphorylated τ protein density via immunohistochemistry, (5) tangles: neurofibrillary tangle counts from silver-stained slides, and (6) Braak stage: a semi-quantitative measure of neurofibrillary tangle pathology[25]. Thus, MD-AD generates six highly related predictions simultaneously and covers each of the two main hallmarks of AD neuropathology (plaques and tangles) at three levels of granularity. The three studies measure partially overlapping subsets of the six neuropathological phenotypes described above (Fig. 1e, Fig. 1a, Supplementary Figure 1a and Supplementary Data 1–2), so across our combined dataset some variables are sparsely labeled, although Braak and CERAD are each measured in all studies (Fig. 1e). During

training, the MD-AD model continually updates model parameters via backpropagation, but only for labeled phenotypes from a given sample. Thus, for each phenotype for a given sample, MD-AD updates parameters from associated separate layers along with all shared layers. This lets us train a unified model from all available samples despite having many missing labels. Although in our application neuropathological phenotypes overlapped across datasets, MD-AD could accommodate non-overlapping phenotypes from different cohorts (as long as they are believed to be closely related and share a common underlying gene expression basis). Details of the MD-AD framework, modeling assumptions, and hyperparameter tuning are provided in Methods.

**MD-AD accurately predicts neuropathology from gene expression, and its predictions are generalizable to external datasets.** In the first pass at model evaluation, we trained MD-AD using standard five-fold cross-validation (CV), and assessed the average $1-R^2_{CV}$ error (mean squared error divided by the phenotype's variance in the test set) on the held-out test samples (Fig. 2a). Our hyperparameter tuning and evaluation procedures are described in detail in the Methods and Supplementary Figure 1e. We compared MD-AD with two simpler baseline models: a regularized linear model (ridge regression) and a single output deep neural network (MLP). These alternative results helped us assess two significant components of the MD-AD model: (1) its non-linear modeling of the relationship between gene expression and neuropathological phenotypes, and (2) its joint modeling of multiple related neuropathological phenotypes. In general, MLP models outperformed linear models, highlighting the advantage of deep learning over a linear approach. Furthermore, compared with the MLP models, MD-AD reduced the prediction error by 7% for CERAD score, 13% for Braak stage, 7% for NPs, 25% for tangles, 10% for Aβ IHC, and 14% for τ IHC (Fig. 2a). Interestingly, MD-AD showed its largest performance gain for the tangles variable, which also had the most missing labels (Fig. 1e), highlighting a specific advantage of joint learning for sparsely labeled data. We additionally experimented with some alternative approaches (e.g., different training/test splits, covariate-corrected data) and found that performance results were robust to these changes (Methods).

Because our model was trained and evaluated on ACT, MSBB, and ROSMAP datasets, we assessed whether residual (uncorrected) batch effects affected performance. To do so, we performed additional validation experiments by leaving out specific datasets during training and then evaluating their performance for MD-AD trained on the other datasets (Fig. 2b, Supplementary Figure 2a). We evaluated the prediction error for ROSMAP alone since it was the only dataset with all six phenotype labels; further, by evaluating a single dataset's performance, we can identify the influence of adding "external" data. We make several observations from this analysis. First, as one may expect, larger training samples always helped to reduce prediction error on test samples from the unseen study (ROSMAP), and especially so when datasets from multiple cohorts were included in the training (i.e., ACT and MSBB) (circular markers in Fig. 2b). Second, when considering the effects of augmenting ROSMAP data with other datasets during training (diamond markers in Fig. 2b), we observed that errors initially increased when adding a new dataset but tended to decline as more datasets were included in the training. This may result from small differences in labeling conventions across studies or batch effects in gene expression data. However, we find that the benefits of additional heterogeneous samples ultimately outweigh potential batch effects in prediction performance. Third, we observed

that adding new samples improved performance for a neuropathological phenotype even when the phenotype in question was not measured in the new samples (see gray footprints around markers in Fig. 2b). The same analysis repeated with the other two cohorts as test sets revealed similar findings (Methods, Supplementary Figure 3). This suggests that the shared representation learned by MD-AD (which is improved by access to additional sparsely labeled samples) captures the underlying biological signal common across noisy neuropathological phenotype measurements.

Next, as the ultimate test of MD-AD out-of-sample predictions, we assessed performance on three independent studies never seen by the model: Mount Sinai Brain Bank Microarray (MSBB-M; $N = 1047$; 565 AD cases and 482 controls), Harvard Brain Tissue Resource Center (HBTRC; $N = 338$; 246 AD cases and 92 controls)[7], and Mayo Clinic Brain Bank ($N = 157$; 81 AD cases and 76 controls)[26]. Because these datasets provide a sparse set of neuropathological labels, we evaluated whether MD-AD predictions were consistent with the (binary) neuropathological diagnosis of AD by calculating "MD-AD neuropathology scores" for each sample (by averaging ranked predictions across the six neuropathological phenotypes).

As shown in Fig. 2c, we observed a highly significant difference in predicted neuropathology scores between AD cases and controls (two-sided $t$ test: $t = 22.98$, $p < 0.001$), and these differences were more pronounced for MD-AD compared with the other baseline models (results split by dataset are shown in Supplementary Figure 4a). More convincingly, when split by age group (Fig. 2c right panel), we consistently observed a significant increase in predicted neuropathology for AD vs control samples, but the difference was largest in individuals under 75 (between groups $p$ values are shown in Supplementary Figure 4b. The same analysis comparing APOE ε4 carriers to non-carriers revealed a similar pattern, shown in Supplementary Figure 5). This is consistent with the observation that aging individuals who are cognitively non-impaired often have substantial neuropathology[21]. Together, these results indicate that MD-AD can identify generalizable gene expression patterns that are predictive of AD-related neuropathology across varied age ranges, and thus it is unlikely that these patterns merely capture normal aging.

**Complex transcriptomic predictors of neuropathology are conserved across species.** We next evaluated how well MD-AD's learned expression patterns predictive of neuropathology recapitulated neuropathology in mouse models. We applied MD-AD trained on human data to make predictions based on 30 brains (hippocampal and cortical) gene expression samples from TASTPM mice that harbored a double transgenic mutation in *APP* and *PSEN1* and compared the predictions to those for 76 samples from wild-type mice[27,28]. We focused on TASTPM mice since they were found to robustly exhibit early signs of amyloid aggregation and plaque formation. As above, to simplify MD-AD predictions, we then predicted all six neuropathological phenotypes via MD-AD and generated an aggregate "neuropathology score" per mouse sample (as described in Methods).

As shown in Fig. 2d, MD-AD predicted significantly higher neuropathology scores for the homozygous cross TASTPM than wild-type mice (two-sided $t$ test: $t = 3.45$, $p < 0.001$). The MLP baseline method also produced significant differences between homozygous and wild-type mice, but less effectively ($t = 3.01$, $p < 0.01$). Furthermore, there was a stronger trend for higher predictions in the heterozygous TASTPM cross samples ($N = 32$) than wild-type mice for MD-AD ($t = 1.38$, $p = 0.17$) compared with MLP baselines ($p = 0.38$). The linear baseline model failed to

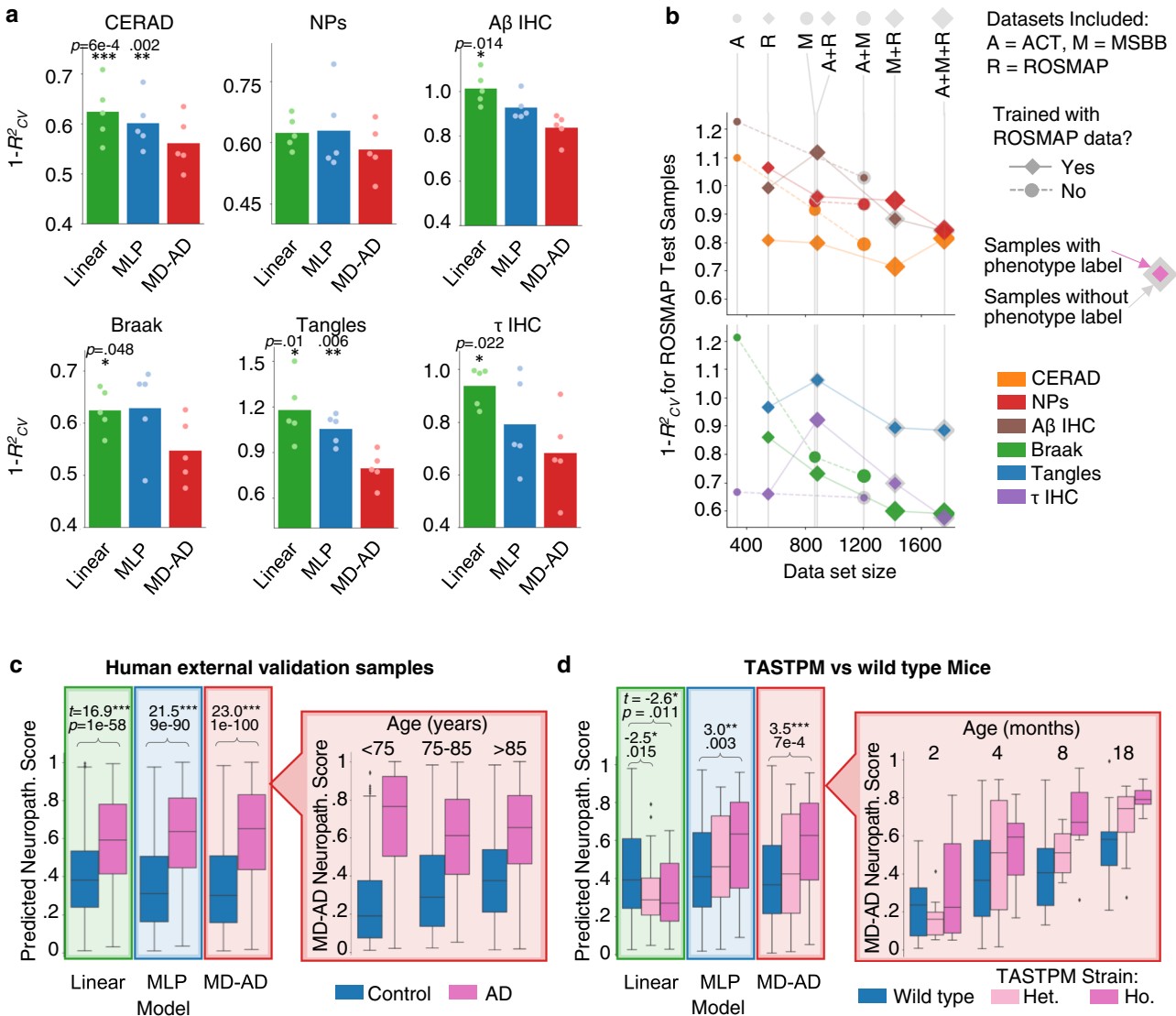

**Fig. 2 MD-AD prediction performance for within-sample test data and out-of-sample (external validation) data. a** Average test set 1-$R^2_{CV}$ (prediction error metric calculated by dividing the mean squared error by the label's variance) for phenotype predictions across five test splits (values from each run as dots). MLP: multiple layer perceptron. Linear: linear model using L2 regularization (two-sided paired $t$ test comparing alternative methods with MD-AD, $p$ values: *<0.05, **<0.01, ***<0.001; $n = 5$ runs of each method). **b** Average 1-$R^2_{CV}$ for ROSMAP test set samples when training on subsets of the available datasets in the training set. **c** For samples from three external validation datasets, we obtain neuropathology scores for each sample from each model. Left: box plots displaying the distribution of predicted neuropathology scores from each method. $T$ test highlight between-group differences for each method (two-sided independent $t$ test, ***$p < 0.001$). Right: box plots displaying the distribution of MD-AD's predicted neuropathology scores split by age group and diagnosis (see Supplementary Figure 4b for sample sizes broken down by age and diagnosis and significance of pair-wise differences). Test statistics were calculated based on 565 AD cases and 482 controls. All box plots in this figure indicate median (center line), upper and lower quartiles (box limits), 1.5 × interquartile range from quartiles (whiskers), and outliers (points). **d** Left: box plots displaying the distribution of predicted neuropathology scores from each method for wild type and TASTPM (both heterozygous and homozygous) mice. $T$ tests highlight between-group differences for each method (two-sided $t$ test, $p$ values: *<0.05, **<0.01, ***<0.001). Right: box plots displaying the distribution of MD-AD's predicted neuropathology scores for mice split by age and strain (See Supplementary Figure 4c for sample sizes and significance of pair-wise differences). Test statistics were calculated based on 72 wild type, 32 heterozygous TASTPM, and 30 homozygous TASTPM mice (same box plot elements as described in part b).

make accurate predictions. None of the models produced significantly different neuropathology scores between other strains (i.e., TPM, TAS10, Tau) and wild-type mice, consistent with lower neuropathological burden in these models (Supplementary Figure 4e). Notably, when we stratified the samples by age, we found that MD-AD tended to predict higher neuropathology in older mice (regardless of strain), but in particular, it made higher neuropathology predictions for homozygous than heterozygous crosses followed by wild-type mice (many of these groups differed significantly from one another, as shown in

Supplementary Figure 4c). Overall, these results indicate that MD-AD learns a generalizable expression pattern associated with neuropathology that is conserved across species.

**Deep transcriptomic signatures of neuropathology are predictive of AD dementia.** Hidden layers of a deep neural network capture the embedding of input examples in the derived feature space, yielding a "hidden" representation that is predictive of the outcome(s) of interest. In this case, the last shared layer of MD-AD (Fig. 1a, c) captures a latent (lower) dimensional

representation of gene expression that is predictive of multiple types of neuropathology related to AD. To derive the biological basis of MD-AD predictions, we first visualized this embedding space in 2D using the t-SNE algorithm (Fig. 3a)[29] (to improve stability, we used a consensus approach over many re-trainings of the MD-AD model, Supplementary Figure 6a). We observed that the representation in this space was impressively coherent with respect to all six neuropathological variables: individuals with similar overall neuropathology severities had similar MD-AD consensus representations for their gene expression profiles, and this observation was true for external test samples not used for model training (Fig. 3d, e, Supplementary Figure 4d). This was remarkable because representations derived by unsupervised dimensionality reduction (e.g., K-means or principal component analysis (PCA)) failed to capture the components of gene expression relevant to neuropathology, and mainly captured effects relatable to batch or brain region differences, whereas those derived by standard single-output MLP tended to overfit to each neuropathology variable and were incoherent *across* neuropathological measurements (Fig. 3c and Supplementary Figure 7).

Next, we evaluated whether the MD-AD embedding can go beyond neuropathology to also capture the molecular manifestation of AD dementia. In particular, we considered three "higher-level" clinical phenotypes: AD dementia (a clinical diagnosis of AD), assessment of cognitive function, and assessment of AD duration. We then correlated the latent representation captured by the hidden nodes in the last shared layer with each of these three higher-level phenotypes. As shown in Fig. 3b, we found that MD-AD consistently produced nodes that were significantly correlated with high-level AD phenotypes; using paired *t* tests, these correlations often outperformed nodes from our MLPs and always outperformed unsupervised methods and module-based approaches ($p < 0.05$ after false discovery rate (FDR) correction over nodes). This indicates that MD-AD creates embeddings that most consistently capture the relationship between gene expression and general AD severity. Together, these results show that by jointly predicting several neuropathological phenotypes, the MD-AD framework produces a low dimensional representation of gene expression data that robustly captures a generalizable signature of AD beyond individual neuropathological phenotypes alone. Detailed annotations for MD-AD embedding nodes are provided in Supplementary Data 3 and Supplementary Figure 6b–d.

**MD-AD reveals an interrelationship between sex and immune genes predictive of AD neuropathology.** We next sought to interpret MD-AD's learned parameters to identify the set of genes (and their relationships) that underlie its impressive predictive performance. Integrated Gradients (IG)[18], one of the most widely used interpretability methods developed for deep neural networks, estimates the importance of input features on a model's predicted output for a particular input sample (See Methods for details). Here, we applied the IG algorithm on the fully trained model in an ensemble fashion to ensure robustness (Methods, Supplementary Figure 8), producing an "importance score" for each gene (Supplementary Data 4). For a global view, we first performed functional enrichment analysis (GSEA[30,31]) using these importance scores (aggregated across samples) and found that relevant genes for the MD-AD model were enriched for several pathways, including the metabolism of RNA and proteins, immune system, cell-to-cell communication, and signal transduction (Fig. 4b). Figure 4a shows the top 50 genes and their pathway annotations where the particular relevance of immune function is even more prominent.

We next assessed to what extent the learned gene importance varied between a linear model and a non-linear model like MD-AD. With a simple linear correlation-based gene ranking (Methods), we found that the top 50 genes were less likely to be annotated to REACTOME pathways (Supplementary Figure 9a). When we directly compared the top 1% of genes from MD-AD versus a correlation-based approach in Fig. 4c, we observed that many genes belonging to metabolism, immune system, and signal transduction pathways were highly ranked for MD-AD but not for correlation-ranking. In contrast, transcription-related genes were more frequently highly ranked for correlation-based rankings compared with MD-AD's rankings. Overall, gene importance scores generated via correlations alone were enriched for more REACTOME pathways (Supplementary Figure 9b), whereas MD-AD offered a more specific set of processes for further investigation (Fig. 5b). We saw similar results when performing the same analyses with KEGG pathways (Supplementary Figure 10)[32].

The non-linear relationships identified by MD-AD can implicitly capture interaction effects with other covariates observable from expression data (e.g., sex, age, medication intake, etc.). Leveraging the fact that, if our model captures a non-linear effect, then two samples with the same expression level for a single gene could receive different IG ("importance") scores by MD-AD (e.g., Fig. 5d; in contrast, a linear model would have no vertical dispersion), we assessed whether a covariate like sex could explain the discrepancy between expression levels and IG scores. (Sex is a major risk factor in AD and has prominent gene expression signatures[33]). In particular, to identify sex-interacting genes relevant to AD, we modeled each gene's per-sample IG score as a linear combination of the gene's expression, the individual's sex, and the interaction between them. Of the 14,591 genes in our dataset, 6465 showed differential MD-AD importance between sexes in an interaction model ($p < 0.05$ after FDR correction), demonstrating that sex-specific expression effects in AD may be widespread. When focusing on the top 100 genes with the highest MD-AD scores, we consistently observed high degrees of interaction between sex and immune system genes (as well as reproduction and hemostasis-related genes) (Fig. 5a, b; we saw similar patterns for KEGG pathways in Supplementary Figure 11b, c). To confirm that genes are not sex-differential by chance, we show the distribution of sex-differential genes compared with the same analysis conducted with shuffled sex labels (Supplementary Figure 11a).

We next explored specific examples of genes with high MD-AD rankings and strong interactions with sex (i.e., the six genes from the top 100 MD-AD list with the strongest interaction *p* values; Fig. 5c, d): *KNSTRN, C4B, CMTM4, TREM2, P2RY11*, and *SERPINA3*. For each of these genes, we observed high expression values associated with higher neuropathology predictions but some stratification across sexes: high expression in females led to especially high neuropathology predictions for *KNSTRN* and *P2RY11*, while the opposite was true for the other four genes. Our finding that immune genes display sex-differential contributions to MD-AD scores appears to be consistent with conclusions from recent studies about sex differences in neuroinflammatory activity and the role these differences may play in neurodegenerative disorders[34].

We note that some of our top sex-interacting genes may play important roles in immune response, particularly in microglia. *TREM2*, which is genetically implicated in AD, interacts with *CD33* (another AD susceptibility gene)[35], is an important contributor in the clearance of toxic Amyloid-β by microglia in mice[36], and is correlated with Aβ deposition in the human brain[35]. Similarly, *KNSTRN* is known to be upregulated in mouse microglial cells' early response to neurodegeneration[37]. These

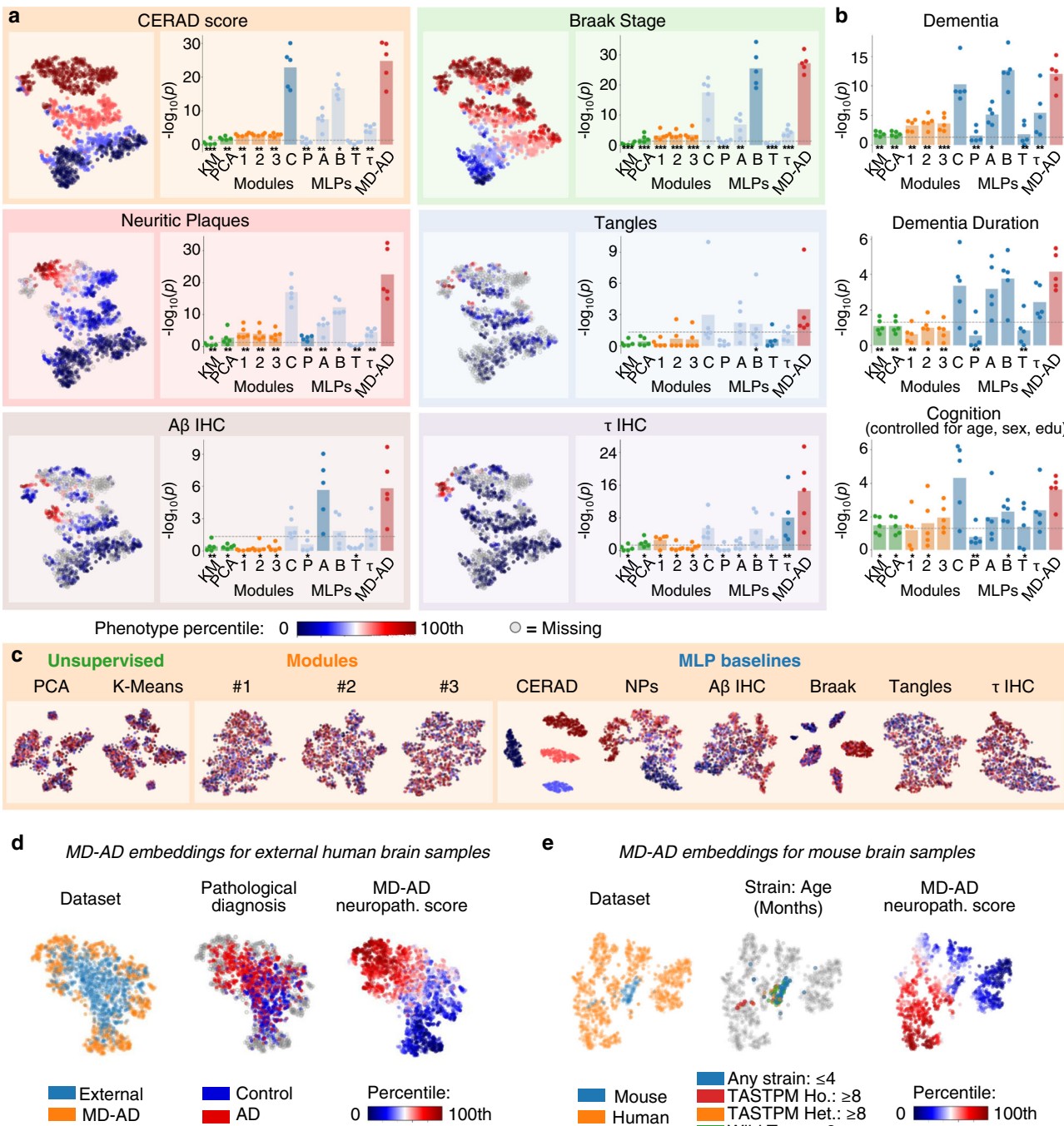

**Fig. 3 Comparing MD-AD's supervised embedding to other embedding methods. a** For each colored box, Left: two-dimensional t-SNE embedding of MD-AD's last shared layer colored by neuropathological phenotype indicated in the title of the box, Right: $-\log_{10}(p\ value)$ of correlations (averaged across five folds) between "best" node from each embedding method and the neuropathological phenotype across 5 test folds. The "best" node was identified as the most significantly correlated in the training set, but bar height indicates the Pearson correlation $-\log_{10}(p$-value) of the node with the phenotype in their corresponding test sets after FDR correction across nodes (averaged results over five runs; individual points show $-\log10(p\ value)$ from each run). Bar graph columns (left to right): two unsupervised embeddings (green; K-Means and PCA), three module-based embeddings (orange; Modules #1[7], Modules #2[6], and Modules #3[9]), six singly trained MLPs (blue), and MD-AD (red). Results from each method were compared with MD-AD using two-sided paired $t$ tests ($p$ values indicated below each bar: *<0.05, **<0.01, ***<0.001). **b** Highest correlation $-\log_{10}(p$ values) (averaged across five folds) found between each embedding method and high-level AD variables: dementia (diagnosis prior to death), dementia duration (approximate time between dementia diagnosis and death; available for ACT and ROSMAP), and last available cognition score (controlling for age, sex and education; available for ROSMAP only). All $p$ values listed are shown after FDR correction over the nodes within each method. Bar height indicates the mean over five folds, and points show individual values from each run. **c** Two-dimensional t-SNE embedding of alternative embedding methods (described in **a**), colored by CERAD scores associated with each sample. **d** Two-dimensional t-SNE embeddings of MD-AD embeddings for training and external datasets. Each point represents a sample colored by dataset (left), AD status for external samples (middle), and MD-AD's predicted neuropathology score (right). **e** Two-dimensional t-SNE embeddings of MD-AD embeddings for external human and mouse samples.

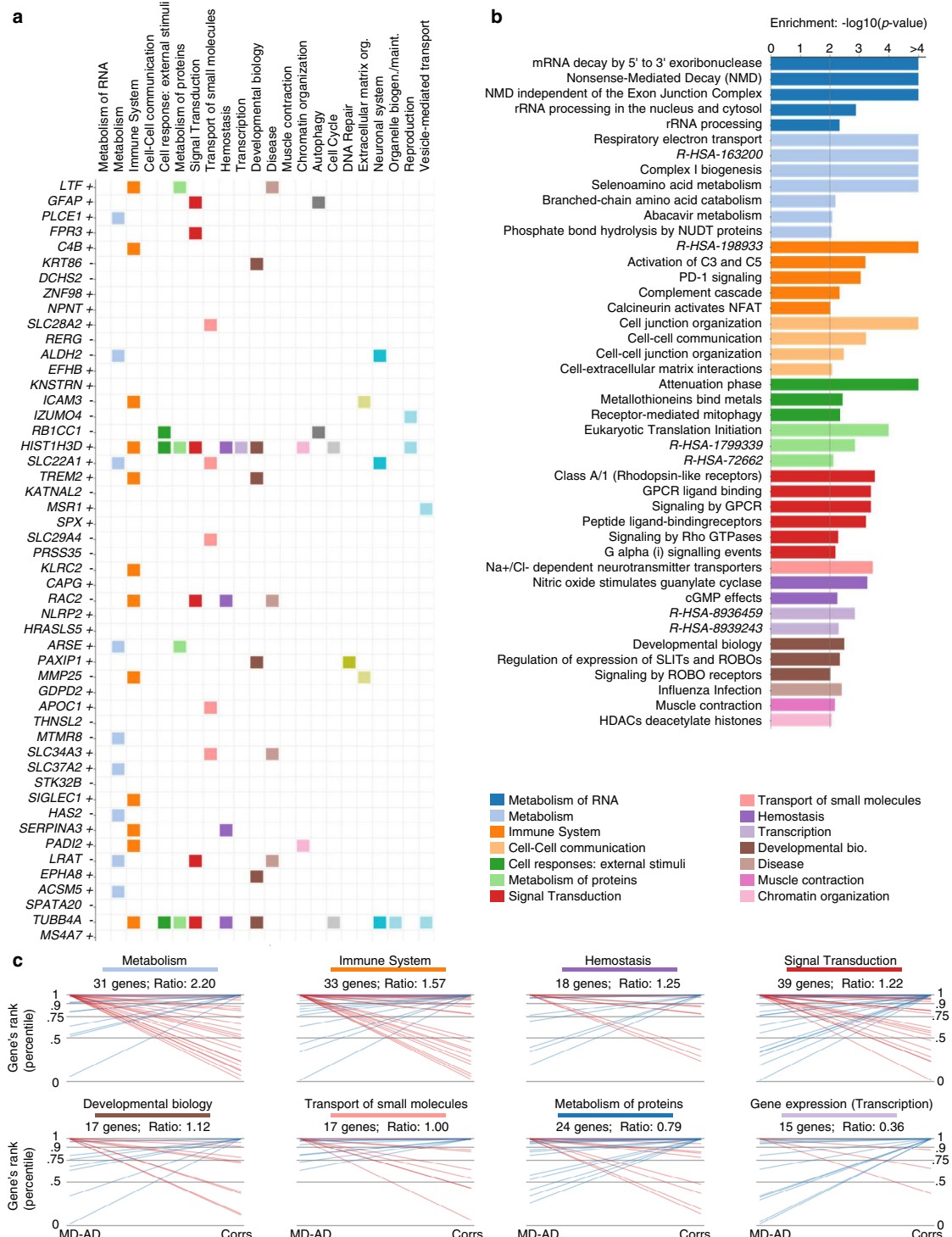

**Fig. 4 Top predictive genes for the consensus MD-AD model. a** Top 50 MD-AD genes and whether they are negatively (−) or positively (+) associated with high neuropathology. Colored squares indicate that the gene belongs to at least one pathway in the column-labeled REACTOME category. **b** Gene set enrichment −log10(*p* value) across the final MD-AD gene ranking for REACTOME pathways. Bars are colored by the pathway's REACTOME category. We show all pathways with significant enrichment (*p* < 0.01). REACTOME pathways with long names are indicated by their REACTOME stable IDs. **c** Comparison of top genes from MD-AD vs a linear correlation-based approach. For each ranking method, we identify the top 1% of all genes and check their membership in REACTOME categories. For each REACTOME category with at least 15 genes in the top 1% of MD-AD and/or correlation rankings, we generate the following plot: each line represents a gene, with the left endpoint at the percentile rank for MD-AD and right endpoint at percentile rank for correlations. For clarity, we color the line purple if the gene falls in the top 1% of both MD-AD and correlations, red if it is only in the top 1% of MD-AD, and blue if it is only in the top 1% of correlations. Finally, the title indicates the ratio of MD-AD to correlation-based top genes for the given REACTOME category.

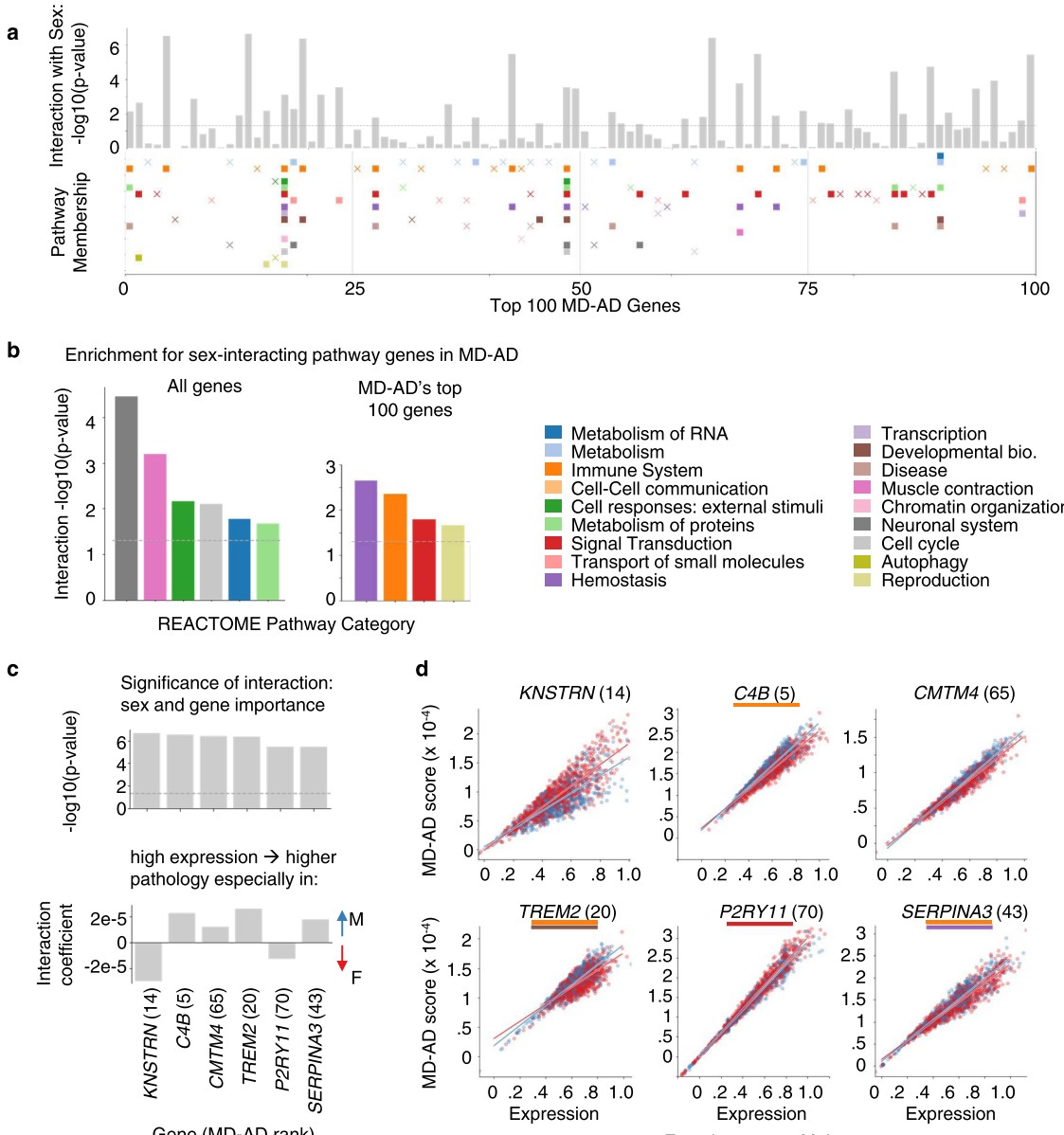

**Fig. 5 MD-AD's top genes and their interactions with sex. a** For the top 100 MD-AD genes, we compute the significance of the interaction between expression and sex for its MD-AD score. The bars indicate the gene's −log10($p$ value) of the interaction term with sex (after FDR correction), and pathway categories each gene belongs to are indicated below. A filled square indicates that the gene significantly interacts with sex ($p < 0.05$ after FDR correction), and an "x" marker indicates that it does not. **b** For genes with significant sex interactions, we compute the significance of the overlap between REACTOME category genes and sex-differential genes among Left: all genes, and right: the top 100 MD-AD genes only. **c** For the top 100 MD-AD genes, we identify the genes with the most significant sex interaction for MD-AD scores. We show the significance of the interaction (top) and the interaction coefficients (bottom) for the top six most sex-differential genes. Each gene's MD-AD rank is indicated in their x axis labels. **d** For the top six most sex-differential top 100 MD-AD genes, we display scatter plots of expression by MD-AD score, coloring each sample by sex of the donor.

findings indicate that MD-AD may capture patterns related to sex-differential microglia activity. To explore this idea further, we obtain lists of upregulated genes from nine clusters of single-cell microglial transcriptomes[38], and compare them to our MD-AD gene rankings. As expected, many top MD-AD genes are upregulated in multiple microglial clusters (Fig. 6a); correlation-based methods ranked these microglial genes less highly (Supplementary Figure 11d). Furthermore, genes upregulated in clusters related to stress, immune function, and proliferation tended to be sex-differential in their gene importance (Fig. 6b), further strengthening the finding that sex differences in immune

response and inflammation may be an important factor in the molecular basis of age-related neuropathology.

To more broadly identify possible cell type-specific effects of MD-AD's important genes, we tested for the enrichment of 41 different cell type clusters (across six cell types) found by single-cell transcriptomic analysis of AD[8]. Here, we found an enrichment of two different microglia clusters, as well as astrocytes and inhibitory neuron clusters (Fig. 6c). Hence, MD-AD's predictions of neuropathology rely on broader transcriptomic events beyond microglia genes, suggesting heterogeneity in the underlying molecular biology that is predictive of accumulation of AD-related neuropathology.

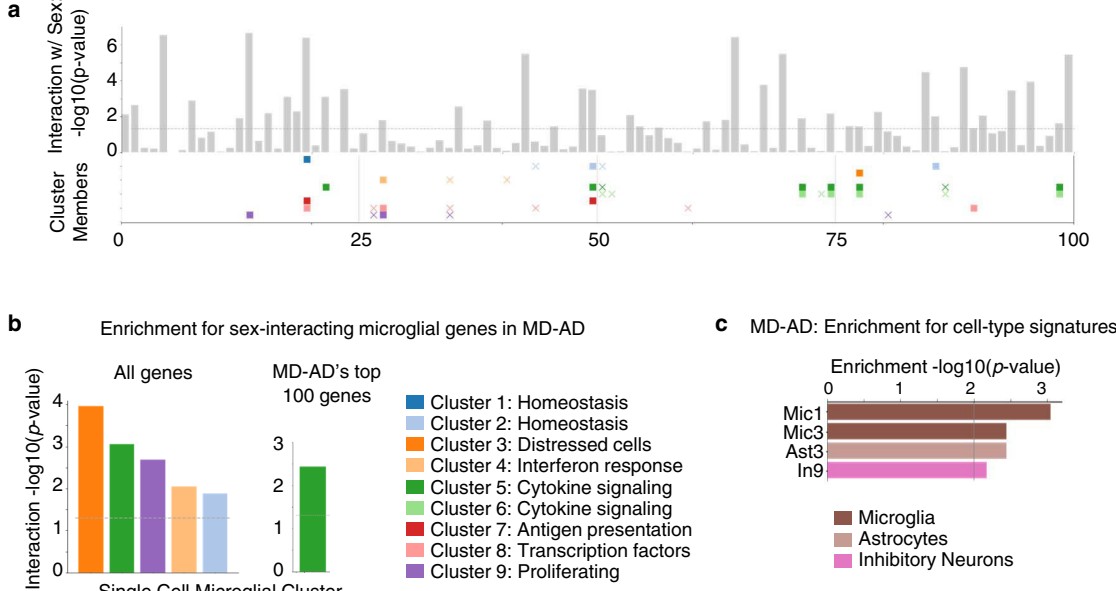

**Fig. 6 MD-AD's reliance on microglial cluster genes and cell type signatures. a** Bars indicate the gene's −log10($p$ value) of the interaction term with sex (after FDR correction), and gene membership in microglial cluster gene sets from Olah et al.[38] is indicated below. A filled square indicates that the gene significantly interacts with sex ($p < 0.05$ after FDR correction), and an "x" marker indicates that it does not. **b** For genes with significant sex interactions, we compute the significance of the overlap between microglial cluster genes and sex-differential genes among: left: all genes, and right: the top 100 MD-AD genes only. **c** Gene set enrichment −log10($p$ value) across the final MD-AD gene ranking for cell type signatures[8].

**Complex transcriptomic predictors learned by MD-AD are conserved across tissues.** Although MD-AD was developed for brain gene expression data, we next asked whether the learned transcriptomic signatures generalize to blood. To this end, we applied our brain-trained MD-AD model to gene expression datasets from two batches of the AddNeuroMed cohort, which we called Blood1 and Blood2 ($n = 711$; NCBI GEO database accessions GSE63060 and GSE63061, respectively; summarized in Supplementary Data 5)[39,40]. As shown in Fig. 7a, MD-AD predicted significantly higher neuropathology scores for individuals with both mild cognitive impairment (MCI) (two-sided $t$ test: $t = 7.34$, $p < 0.001$) and AD dementia (two-sided $t$ test: $t = 5.87$, $p < 0.01$) compared with cognitively normal controls (CTL). Consistent with external brain samples shown in Figs. 2d and 2f, MD-AD predictions tended to increase with age for cognitively normal individuals, while they were consistently significantly higher for MCI and AD individuals compared to controls for individuals under 80 years old (Fig. 7b, Supplementary Figure 12b). Importantly, we noted that a linear model failed to make meaningful predictions (Fig. 7a and Supplementary Figure 12a), suggesting that complex models like MD-AD have better performance in extracting the true underlying signal transferrable between tissues than linear models.

Next, we evaluated whether the patterns captured by the MD-AD model were consistent across training brain gene expression samples and blood. To this end, we again visualized MD-AD's learned embedding using the t-SNE algorithm (Fig. 7c). We noted a clear difference in expression patterns between blood and brain samples (as seen by the clustering of blood samples in Fig. 7c); however, MD-AD nevertheless produced an embedding for blood data that stratified blood samples along predicted neuropathological phenotypes in a manner highly consistent with the blood donor's cognitive status (Fig. 7c; Supplementary Figure 12c). Together, these analyses indicate that jointly learning the relationship among brain gene expression and several neuropathological phenotypes may allow for learned representations that span tissues. This in turn can open avenues for early identification of individuals at risk, and provide clues into tissue-agnostic molecular mechanisms underlying AD dementia.

## Discussion

We introduce MD-AD, a deep neural network approach for jointly modeling the relationship between brain gene expression and multiple sparsely labeled neuropathological phenotypes in a multi-cohort setting. By exploiting the synergy between deep learning and a multi-cohort, multi-task setting, we demonstrated that MD-AD can capture complex, non-linear feature representations that are not learned using conventional expression data analysis methods. Specifically, we observed that multi-task learning improves prediction performance over single-task models. Adding data from different cohorts improves performance for various neuropathological phenotypes, even those that lacked labels. When we extended our method to other datasets, it captured AD-related biological signals, showing that MD-AD can transfer effectively to out-of-cohort, out-of-species (mouse), and even out-of-tissue (blood) datasets.

As a neural network framework, MD-AD's last shared layer embedding reveals high-level features of gene expression that are predictive of neuropathology according to the intermediate components of the model. As expected, owing to multi-task supervision, our embedding nodes tend to relate to AD-associated neuropathology far more effectively than do standard unsupervised approaches and earlier reported (unsupervised) module-based approaches. Compared with single task-supervised neural networks, MD-AD's joint training consistently provided a more stable and coherent AD-related embedding. By exploring the molecular pathways relevant to each node, we identified relevant gene sets contributing to these high-level AD-related features of gene expression.

Finally, we leveraged the complex relationships learned by MD-AD to refine our understanding of the molecular drivers of AD neuropathology. By interpreting genes relevant to our model's predictions, we uncovered that MD-AD relied on many genes not found in earlier linear-based methods, including several

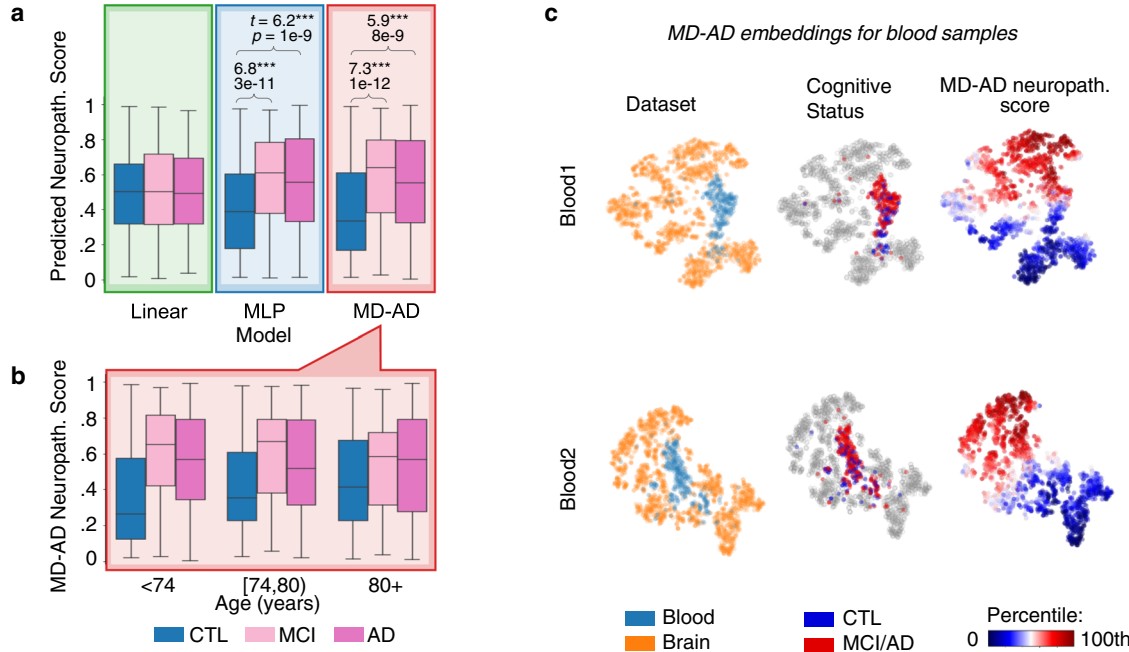

**Fig. 7 MD-AD's transfer performance for blood gene expression datasets. a** Box plots show neuropathology predictions from each method split by cognitive status. From left to right, individuals who are cognitively normal (CTL), have mild cognitive impairment (MCI), and Alzheimer's dementia (AD). *T* tests highlight significantly different groups within each method (two-sided *t* test, ***$p < 0.001$). All box plots in this figure indicate median (center line), upper and lower quartiles (box limits), 1.5× interquartile range from quartiles (whiskers), and outliers (points). $N = 238$ CTL, 189 MCI, 284 AD samples. **b** Box plots show the differences in MD-AD predicted neuropathology samples from individuals stratified by age group and cognitive status (significant differences are shown in Supplementary Figure 12b; $n = 238$ CTL, 189 MCI, 284 AD samples) (same box plot elements as described in part a). **c** t-SNE embedding of last shared layer from MD-AD models trained for Blood1 and Blood2 datasets. Samples are colored by their dataset (left), cognitive status (while brain samples are shown in gray; middle), and predicted neuropathology score (right).

immune genes. These findings expand the general narrative established by human genetic studies of AD and now a proteomic study of AD;[41] in particular, we see enrichment for complement pathway genes (Fig. 4), which likely connect with the role of the complement receptor 1 (*CR1*) gene, which harbors an AD susceptibility variant whose functional consequences remain poorly understood but do include an influence on the accumulation of neuritic plaque pathology[42–45]. Thus, MD-AD results converge with human genetic results to emphasize the role of complement in AD; interestingly complement protein C4B emerges as one of the top pathology-related genes that display a strong interaction with sex, with men showing a much stronger association than women (Fig. 5c). This is similar to the behavior of *TREM2*, another well-validated AD susceptibility gene (Fig. 5c); however, its relation to amyloid pathology in ROSMAP data was previously reported as being modest[35]. MD-AD was able to uncover its more prominent role in transcriptional data, which is obscured by its sex-dependent nature. Likewise, women reported to have higher expression of a signature of aged microglia in these data[34], and two modules of co-expressed cortical genes enriched for microglial genes and associated with amyloid (module m114) or tau (module m5) pathology are also influenced by sex[46]. However, the role of neither group of genes is explained by sex; this indicates that the role of sex in the impact of the immune system in AD is complex. MD-AD was able to uncover this complexity more effectively, as is illustrated in Fig. 5c where some genes have greater effects in men and others in women. Thus, it is not the case that role of the immune system is polarized in one of the two sexes; rather, some pathways and perhaps certain cell subsets may have a larger role in women while others are dysfunctional in men. This could explain why the role of immune genes is more prominent in our analyses: reports from simpler linear models

often included immune pathways[6,7,47,48] but other pathways usually figured more prominently in these earlier RNA-based network models. A meta-analysis of RNA studies (which include the ROSMAP data) highlighted the larger number of sex-influenced genes among the AD-associated gene modules and noted that microglial cells appear to be enriched for both male and female-specific expression effects. We note that like any other machine learning-based model applied to observational data, we are unable to directly infer causality in our framework, as both gene expression and neuropathology data used for MD-AD were collected from postmortem brains. Nevertheless, with our list of results and our careful evaluation of sex effects we now have an important road map with which to guide our exploration of the role of microglia in AD in a sex-informed manner. This perspective will be critical not only for mechanistic studies whose results could be obscured by sex effects but also, more importantly, by guiding the study design of clinical trials as highly targeted therapeutic agents emerge to modulate the immune system in AD.

This is but one of the narratives that have emerged from our initial deployment of the MD-AD approach in the aging brain. As new cohorts are characterized, sample sizes expand and new data such as single-nucleus RNA-sequencing profiles emerge, our approach will help to facilitate data integration and to uncover insights that would not otherwise emerge. Beyond enabling good predictions, our report may actually highlight a more important contribution of MD-AD in resolving key elements of the data structure in the nodes that we defined: these are more than simple aggregates of factors with predictive power. They are beginning to uncover complex interactions, such as the impact of sex, which is involved in both men and women, but in different ways, making it difficult to appreciate the role of certain immune pathways in

simpler statistical models. Beyond producing accurate and generalizable neuropathology predictions and improving our biological understanding of AD pathogenesis, MD-AD provides a framework for integrating and analyzing gene expression data from separate cohorts and identifies common underlying relationships among gene expression and phenotypes of interest, which may be expanded as new data emerge.

## Methods

**Data processing**. For developing the MD-AD model, we used RNA-Seq and neuropathology datasets available through the AMP-AD Knowledge Portal: (1) ACT[21], (2) MSBB[22], and (3) ROSMAP[6,19,20]. Details of sample collection and sequencing methods have been described previously [6,19–22]. We pooled together brain gene expression data from the temporal cortex, parietal cortex, hippocampus, and forebrain white matter from ACT, Brodmann areas 10, 22, 36, and 40 from MSBB, and the dorsolateral prefrontal cortex from ROSMAP. To reduce confounded relationships, we excluded samples from individuals who had neuropathological diagnoses other than AD. Taken together, the studies provide 1758 gene expression samples.

In all three studies, extensive quality control measures were taken during the original processing of the data, as described by the papers introducing transcriptomic datasets for ACT[21], MSBB[22], and ROSMAP[6] cohorts. All samples that passed quality control checks in these individual studies were included in our study. If Ensemble gene IDs were provided, we mapped them to gene symbols to keep consistent gene identifiers across datasets. In order to compile gene expression samples across the three cohorts, we retain expression levels for genes that are present in all datasets. Within each dataset, we exclude genes with null values for over two-thirds of samples. For ACT and ROSMAP, gene expression measurements were provided in normalized FPKM units, so we log-transformed the RNA-Seq datasets to obtain gene expression datasets that were roughly normally distributed (whereas MSBB gene expression data were already normalized). Then, for all datasets, we normalized values such that each gene's expression measures varied between 0 and 1. We then combined the gene expression datasets and kept all 14,591 genes that are present across all three datasets. Of these genes, 96.3% are autosomal (3.5% on only the X chromosome, 0.1% on only the Y chromosome, and 0.1% on both X and Y chromosomes). Finally, we performed batch effect correction with ComBat to reduce systematic differences across studies (Supplementary Figure 1c)[23].

Next, for each gene expression sample, we incorporated the available corresponding neuropathology labels: (1) Aβ IHC: amyloid-β protein density via immunohistochemistry, (2) plaques: neuritic amyloid plaque counts from stained slides, and (3) CERAD score: a semi-quantitative measure of neuritic plaque severity[24], (4) τ IHC: abnormally phosphorylated τ protein density via immunohistochemistry, (5) tangles: neurofibrillary tangle counts from silver-stained slides, and (6) Braak stage: a semi-quantitative measure of neurofibrillary tangle pathology[25]. Detailed descriptions for each neuropathological phenotype within each dataset are provided in Supplementary Data 1. Because Braak stage and CERAD score are global measurements of neuropathological damage, if an individual had multiple available gene expression measurements from different regions, each sample was labeled with the same Braak and CERAD values. However, Aβ-IHC and τ-IHC were provided for several brain regions for both ROSMAP and ACT studies. Therefore, each expression sample was labeled with the Aβ-IHC and τ-IHC measurements for the same or nearest region. Because the available plaques label provided by MSBB was averaged over several brain regions, we similarly used ROSMAP's average plaques and tangles labels (aggregated from several regions) for consistency with MSBB's metrics. We provide demographic and neuropathology information about individuals in each cohort in Supplementary Data 2. Finally, for consistency across datasets, we first normalized all neuropathological variables to vary between 0 and 1 before combing datasets.

**Computational methods: review of previous approaches**. Post-mortem transcriptomic studies have investigated molecular and neuropathological outcomes in AD. Early work in this domain examined simple correlations among gene expression and AD symptoms[10] or compared gene expression levels across AD patients versus controls[11]. More recently, systematic network-based analyses have contributed to the understanding of AD biology. In particular, Zhang et al.[7] constructed molecular networks based on bulk gene expression dataseparately for individuals with and without AD and identified modules with remodeling effects in the AD network. Mostafavi et al.[6] used co-expressed genes in the aging human frontal cortex to build a single molecular network and identified modules related to AD neuropathological and cognitive endophenotypes. Using single-cell RNA-sequencing data, Mathys et al.[8] clustered cells within brain cell types to identify and characterize AD-related cellular subpopulations. Each of these approaches has been applied to single cohorts. Until recently, unified and robust modeling of AD neuropathology based on brain gene expression has been hindered by relative scarcity and regional heterogeneity of brain gene expression datasets. One possible solution is to combine multiple datasets to gain statistical power. The collection of

postmortem brain RNA-sequencing datasets, assembled by the AMP-AD consortium, provides new opportunities to combine multiple datasets. However, such heterogeneous datasets pose challenges to many methods, which must account for inter-study differences. In a recent attempt, Logsdon et al.[9] used a meta-analysis approach to identify co-expressed modules separately for seven brain regions across three datasets, then subsequently applied consensus methods to identify modules that were conserved across multiple regions and studies. As of now, we're not aware of any methods that directly model all data in a unified way.

**Modeling assumptions**. For modeling gene expression and neuropathology data, we make several assumptions.

*Common assumptions of machine learning models.* First, regarding the stability of our models, we assume that a single training of each model is representative of all training instances. In order to buffer the potential failure of the assumption, we used to select hyperparameters across different splits of our data across MD-AD, singly trained MLPs, and linear models. Further, our final model is trained 100 times to generate ensemble predictions and interpretations, as described in future sections. Second, we assume that our samples are "sufficiently" independently and identically distributed (i.i.d.) such that a model trained on these samples should generalize well to new samples from the population of interest. Third, we assume that the true data distribution is smooth such that samples with very similar gene expression values should display similar neuropathology.

*Additional assumptions of MD-AD.* Deep learning relies on the assumption that the data is generated by a composition of (learnable) features in a hierarchical manner. This allows neural networks with multiple layers to collapse correlation patterns in the input space to generate intermediate embeddings in a way that is useful for prediction[14]. Unlike linear models, our deep learning framework does not assume that there's a linear relationship between the predictors and outcomes, nor does it require normally distributed predictors, or low multicollinearity. Although deep learning relies on relatively few assumptions, in practice, some of these assumptions do not fully hold. In particular, our samples are certainly not i.i.d., as some samples are derived from the same brain. Thus, external validation is invaluable for evaluating the effectiveness of our framework in new settings with no information leakage. The observation that MD-AD transfers well to separate datasets (and even species and tissues) implies that our framework is effective regardless of whether these assumptions were fully upheld. Finally, multi-task modeling frameworks hinge on the assumption that there are shared common information across neuropathological phenotypes[13]. In combining multiple datasets with different sparsity patterns, we additionally assume that this common representation is consistent across cohorts and is generalizable to new datasets. This assumption appears to hold, as demonstrated by the improved test performance of the multi-task network over singly trained MLPs, and improved ability to generalize well to external datasets.

**The MD-AD model**. MD-AD, is a unified framework for analyzing heterogeneous AD datasets to improve our understanding of the expression basis for AD neuropathology (Fig. 1). Unlike previous approaches, MD-AD learns a single neural network by jointly modeling multiple neuropathological measures of AD severity phenotypes, and hence can incorporate data collected from multiple datasets. This unified framework has key advantages over separately trained models. First, MD-AD allows sparsely labeled data, which is a natural characteristic of datasets aggregated through consortium efforts (Fig. 1e). Even if different phenotypes only partially overlap in the measured samples, each sample contributes to the training of both phenotype-specific and shared layers. Predicting multiple phenotypes at once biases shared network layers to capture relevant features of these AD phenotypes at the same time. This is of critical importance: each phenotype represents a different noisy measurement of the same underlying true biological process, and as we demonstrate by joint training MD-AD is able to average out the noise to extract the true hidden signal. In addition, the increased sample size enables MD-AD to capture complex non-linear interactions between genes and phenotypes. In contrast, MLPs offer another powerful approach for directly capturing complex relations between gene expression and a neuropathological phenotype. However, training separate MLPs for each phenotype (Supplementary Figure 1a) has limited scope: it can utilize only the samples measured for a specific phenotype, and it cannot share information across related phenotypes. We demonstrate that these advantages improve MD-AD prediction accuracy, enabling predictions to generalize across species and tissue types (Fig. 1b). As illustrated in Fig. 1a, the MD-AD network jointly predicts six neuropathological phenotypes from gene expression input data via shared hidden layers followed by task-specific hidden layers.

**Training and evaluating MD-AD**

*Pre-processing with PCA.* In order to have efficient and robust training and to reduce overfitting, we apply a PCA transformation to the data and use the resulting top 500 principal components—a 500-dimensional representation of our 14,591 gene expression values—as the input to the MD-AD and all baseline models. This

approach is consistent with the use of PCA for pre-processing in other studies that have employed deep learning in gene expression analyses[49–51]. Our choice to use 500 PCs is supported by some preliminary analyses of AD-related signals captured by various PCs. First, as shown in Supplementary Figure 1b, the cumulative variance explained by 500 PCs is 92%, indicating that reducing our input features by a factor of about 30 still retains most of the variation in the data. However, we note that by using only 500 PCs, we may lose some information that may be especially predictive of AD neuropathology. To investigate this potential issue, we sought to predict average neuropathology scores from different sets of PCs using a linear model, and compare predictive performance to the full set of genes. We use CV to tune the alpha parameter and, based on the same training/testing splits used in our main analyses (described below), find that by the time we include up to 500 PCs, we have reached similar predictive performance between a model trained on PCs versus the raw gene expression features (Supplementary Figure 1d). This suggests that the linear transformation provided by the first 500 PCs retains features of gene expression data that are almost as linearly predictive of AD neuropathology as the full dataset using all genes.

*Construction of models.* For comparison to MD-AD, we generate six analogous MLP networks with un-shared representations, and six linear models containing no hidden layers, to serve as baseline models (see Supplementary Figure 1a). All models were built using Tensorflow and Keras packages, and were constructed as consistently as possible, with the same inputs. The MLP baseline model was identical to the MD-AD model except with only a single branch of task-specific layers. Similarly, the linear baseline models were identical to the MLP baselines but with all hidden layers removed. Each model was trained on a single Nvidia GeForce GTX 980 Ti GPU. Although training time may vary across machines, we found that training the MD-AD model on the full dataset took about 350 seconds on average.

*Hyperparameters.* After some preliminary experiments with single- and multi-task neural networks, we decided to train all networks with ReLU activations and dropout units (with drop-out rates of 0.1) and trained each model for 200 epochs with batch sizes of 20 using adam optimization. These settings were selected because they led to relatively stable and effective predictions. We tended to see some variation in performance based on kernel regularization, and hyperparameters of the optimization method, so for hyperparameter tuning (described below), we performed grid search over the following hyperparameters: kernel regularization parameter (1e-3 vs 1e-5), gradient clip norm (0.1 vs 0.01) for the adam optimizer, and the learning rate (1e-3 vs 1e-4).

*Cross-validation (CV) and model tuning.* For our model training and evaluation, we use a modified Cross-validation (CV) and testing scheme as illustrated in Supplementary Figure 1e, in which we perform five separate rounds of model tuning with CV followed by evaluation in a test set. For a single round, one-fifth of all samples are assigned to a held-out test set. Then using the remaining 4/5ths of the samples, we perform five-fold CV to select hyperparameters with the best prediction performance. We then train the selected model using the full training set (4/5ths of the original data) and then report performance on the held-out test set. In order to evaluate the robustness of our evaluation metrics under different splits, we initially split the full dataset into five separate groups and repeated the above process five total times, where each one-fifth of the data acted as a held-out test set once. We note that across these iterations, different training sets selected different configurations of hyperparameters, and for each train/test round, we trained the full training set on the specific configuration selected by CV in that training set. Thus, our test set evaluations (e.g., in Fig. 2a) reflect average test performance for the selected models in each round.

For MD-AD, we additionally explored several alternative options for architectures with different amounts of shared and task-specific layers (Supplementary Figure 2b, c). We selected the final architecture (shown in Fig. 1a) because we wanted to have multiple hidden layers in both the shared portion and task-specific portion of the network to allow for non-linear interactions to be learned in both the shared representation and in the task-specific branches. However, when we evaluated alternatives to this approach (using the same selected hyperparameters for our original MD-AD model), we found that alternatives to this approach tended to perform similarly or worse (Supplementary Figure 2b, c).

*Evaluation metrics.* As described above, for each round of train/test splits, we use five-fold CV to make modeling choices for the MD-AD model and baselines before training each model with the full training set and reporting and reporting test $1-R^2_{CV}$ error (mean squared error divided by the phenotype's variance in the validation set; averaged over all five test splits). We evaluate model performance in two ways: (1) standard train and test sets, and (2) ROSMAP test performance for different subsets of the available datasets.

First, separately for each of our five CV training sets, we calculate the final test MSE on the corresponding hold-out set. To test whether these effects are significant, for each baseline method, we performed one-sided paired $t$ tests to determine whether there is a significant difference between the baseline method's error and MD-AD's across the five test folds (Fig. 2a).

Next, in order to evaluate the contributions of each dataset to prediction performance, we performed the above procedure with different subsets of available datasets. Because ROSMAP is the only dataset with all available neuropathological phenotypes, we evaluate performance specifically on ROSMAP. In Fig. 2b, we show ROSMAP test samples' MSE performance when trained on all subsets of ACT, MSBB, and ROSMAP training samples (following the same CV procedure described above). We additionally repeated the same analysis using MSBB and ACT test samples and computed their prediction performance for available phenotypes (Supplementary Figure 3a). In order to evaluate how to transfer performance (i.e., training and evaluating with samples from disjoint cohorts) was impacted by the addition of samples, we performed an additional analysis where we trained with ACT samples and varying fractions of MSBB samples to see how additional MSBB samples impacted ROSMAP test performance (and also evaluated the reverse, training on ROSMAP and MSBB and testing with ACT samples) (Supplementary Figure 3b). Interestingly, we saw that ROSMAP test performance improves with the addition of MSBB samples during training with ACT, whereas ACT samples generally do not improve with the addition of MSBB samples during training with ROSMAP. This may imply that there are more pronounced distributional differences for the ACT cohort when compared with other cohorts, or that improvements are more apparent when the training set is much smaller (as is the case when training with only ACT samples).

*Final model selection.* Finally, after our in-depth CV and testing scheme were used to evaluate our methods internally, we constructed "final models" for external validation and model interpretation. First, we selected a single set of hyperparameters for each model by ranking each configuration's prediction performance for each round and then choosing the configuration with the highest average rank. The selected hyperparameters for each "final model" are provided in Supplementary Data 6. We trained "final models" for MD-AD and baselines by each using a single set of hyperparameters on the full dataset.

*Evaluating models with covariate-corrected data.* Gene expression-related covariates may influence gene expressions in a systematic way, and thus should be critically considered. Indeed, there does seem to be a small but significant correlation between neuropathology scores and both postmortem interval (PMI; $r = -0.16$, $p = 1e-9$) and RNA integrity number (RIN; $r = -0.09$, $p = 0.002$), which are both features which may influence measured gene expression. In our study, we chose to leave our expression profiles uncorrected for all covariates, and instead allow MD-AD to learn from the available gene expression patterns so that we can subsequently assess how these covariates interact within our final models.

Although Supplementary Figure 6c shows that PMI and RIN had modest residual correlations with nodes in the consensus MD-AD network, and thus likely do not appear to be driving forces in our model, we performed an additional analysis to ensure that gene expression-related covariates were not an important factor in our prediction performance results presented via our CV evaluations. To that end, we use the following method to correct our gene expression data for sequencing-related covariates: we linearly regressed PMI and RIN from our expression inputs by modeling the expression of each gene as a linear regression with PMI and RIN. We then saved the residuals of the predicted expression value as our corrected expression values. We then performed the same model training and evaluation procedures as described above using the corrected gene expression values as inputs, and found that these results were quite similar to our original results with uncorrected gene expression values (Supplementary Figure 13a, b). Together, these findings indicate that covariates related to gene expression measurement procedures do not seem to have a large impact on our final results, nor does the MD-AD heavily rely on these covariates, and for that reason, our main results are all based on gene expression data without covariate-correction.

*Evaluating models with fully independent CV splits.* All internal validation results were presented for the same CV and testing splits. We note that in generating these original splits, we randomly assigned all samples within each cohort. However, because ACT and MSBB datasets provide multiple samples (collected from different brain regions) from each individual, there are many individuals in our dataset with samples (of different brain regions) in both training and test splits. In order to ensure that performance improvements seen for MD-AD versus MLP or linear baselines were not due to our splitting choice, we repeated our CV experiments using a new method of splitting samples. Instead of splitting samples completely randomly as was done to generate our main internal test results (i.e., in Supplementary Figure 1e), we instead split individuals randomly for each dataset to ensure that no samples from the same individual could be split across training and validation sets.

We performed the same CV and hyperparameter selection process as was done for our original dataset splits, and our resulting prediction performance and last shared layer evaluations are shown in Supplementary Figure 13c, d. In these experiments, we find that MD-AD (as well as the baseline methods) provides very similar prediction performance when trained and evaluated on fully separated training and test sets, suggesting that our original results did not seem to hinge on the similarity of samples between the training and test set. We similarly find that MD-AD continues to produce embeddings that capture both neuropathology phenotypes and higher-level AD variables more consistently than alternative

approaches. Finally, we note that the hyperparameters selected using the new splits are similar to the original final model selected from our original splits with the exception of a single hyperparameter (kernel regularization of 0.001 for our original splits, compared with 0.00001 for the new splits). However, we find that in our analyses with the new splits, a model trained on our originally selected set of hyperparameters has a very similar performance to the newly selected set of hyperparameters. Together, these results indicate that our choice to split samples randomly produced very findings to the alternative of splitting samples pseudo-randomly by individual.

**External validation: out-of-cohort human brain samples**. In order to evaluate MD-AD's ability to generalize to out-of-sample data, we assessed performance on three datasets: Mount Sinai Brain Bank Microarray (MSBB-M; $N = 1047$), Harvard Brain Tissue Resource Center (HBTRC; $N = 338$), and Mayo Clinic Brain Bank ($N = 157$). These datasets were collected from AMP-AD (with the exception of HBTRC that was collected from GEO: GSE44772) but were left out of the original MD-AD training because they were microarray samples or lacked many neuropathology labels.

After normalizing gene expression samples from external datasets in the same way as described for the ACT, MSBB RNA Seq, and ROSMAP datasets, we then adjust the expression values to have similar distributions to our batch corrected training datasets. We evaluated the MD-AD model on our new processed data to obtain predictions for all six phenotypes. Because these three external datasets provide a sparse set of neuropathological labels, we do not have access to labels for many of the six MD-AD labels. Instead, we evaluated whether MD-AD's predictions were consistent with the (binary) neuropathological diagnosis of AD, by aggregating MD-AD's various neuropathology predictions into one "neuropathology score". The "neuropathology score" was produced by first calculating percentiles across samples (within each dataset) for each neuropathological phenotype, then averaging over the six phenotypes.

Figure 2c shows that MD-AD provides the largest differences in neuropathology scores between individuals with and without neuropathological diagnoses of AD. We further compared neuropathology scores between AD and non-AD individuals split by age group (the significance between groups shown in Supplementary Figure 4b).

A similar analysis was carried out comparing carriers of the APOE ε4 allele with non-carriers (instead of AD vs control individuals). Results shown in Supplementary Figure 5a–c revealed similar patterns, including improved discrimination between groups for MD-AD compared with MLP and linear baselines, and more pronounced differences in predicted neuropathology for younger APOE ε4 carriers versus non-carriers. However, when comparing predicted neuropathology between APOE ε4 carriers versus non-carriers within the same cognitive diagnosis, we do not see a difference in predicted neuropathology for AD-afflicted APOE ε4 carriers and non-carriers (Supplementary Figure 5d, e).

**External validation: mouse samples**. To evaluate how well expression patterns predictive of neuropathology learned by MD-AD recapitulates neuropathology in mouse models. To that end, we obtained gene expression data from Matarin et al.[27] for 15 TASTPM mice that harbor a double transgenic mutation in APP and PSEN1, as well as 37 wild-type mice. For each mouse, brain gene expression was measured from two samples collected from the cortex and hippocampus, doubling the total sample size. Data were quantile-normalized and log-transformed. For this experiment, we mapped mouse to human genes (via gene symbols) for a total of 7057 intersecting genes between our training dataset and the mouse expression data, which were again normalized to follow the same distributions as our MD-AD training data. We re-trained our MD-AD model on only these 7057 genes for all MD-AD samples and then generated "neuropathology scores" for the mouse samples exactly as described in the previous section. As with external validation experiments described above, we compare MD-AD with MLPs and linear models in separating neuropathology scores between TASTPM and wild-type mice (Fig. 2e). We also show differences in neuropathology scores between different age groups (Fig. 2d, Supplementary Figure 4c).

**Validation of supervised embedding**. The output of an intermediate layer of a neural network can be viewed as the lower dimensional embedding of the input features. In this paper, we focus on the last shared layer of the MD-AD network because it is a supervised embedding of gene expression data that is influenced by all six training phenotypes. We evaluate the embedding compared with those generated by both singly trained MLPs as well as unsupervised methods (i.e., K-means and PCA) in two ways: (1) high-level visualization with t-SNE, and (2) evaluating the correspondence between individual nodes and AD-related features.

*Visualizations with t-SNE*. For each of the MD-AD, MLP, and unsupervised models, we train the models on the full combined dataset. For the deep learning models, we then generate "supervised" embeddings by obtaining the output of the last shared layer (or analogous layer of the MLP model). For the unsupervised methods, K-Means and PCA, we generate an embedding of 100 dimensions to be consistent with the MD-AD and MLP models. After generating these embeddings for all samples, we then compress them into two dimensions via the t-SNE

algorithm[29]. T-SNE Visualizations of MD-AD's supervised embedding are shown in Fig. 3a (left side for each phenotype), and the figure is replicated six times, with each plot showing samples colored by neuropathological phenotype severity for each of the six phenotypes. For comparison, t-SNE visualizations for the singly trained MLPs and unsupervised methods are shown in Fig. 3c (colored by CERAD Score only) and colored by other characteristics and covariates of interest in Supplementary Figure 7.

*Node-phenotype correlations*. To test whether MD-AD's embedding generalizes more to AD phenotypes than the alternative methods, we compare the nodes that best capture each phenotype among MD-AD, MLPs, and unsupervised methods. We perform the following analysis with the same five training and test splits described earlier: for each of the six phenotypes used in MD-AD's training, we identify the node in MD-AD's last shared layer whose output is most significantly correlated with that phenotype in the training set. We then report the $-\log10(p$ value) (after FDR correction over nodes) for the correlation between that node's output and the training phenotype in the test set, averaged across the train/test splits. (Fig. 3a, right side for each phenotype).

We also perform a similar analysis with higher-level AD phenotypes not used during model training: dementia diagnosis (binary variable available in all datasets), last available cognition score (controlling for age, sex, and education; only available for the ROSMAP dataset), and AD duration (i.e., the time between dementia diagnosis and death; available for the ACT and ROSMAP datasets). For this analysis, we report the highest $-\log10(p$ value) after FDR correction between nodes and the high-level phenotypes, average over the five test sets (Fig. 3b).

**Model interpretation with Integrated Gradients (IG)**. Although deep learning models have shown promise in biological and health applications, they have been limited by the difficulty of explaining their predictions. Fortunately, the development of interpretability methods for "black box" models such as deep neural networks have helped researchers derive understanding from complex models[52]. In particular, IG is a method for assigning sample-specific importance scores for inputs of a model on the output based on the gradients of neurons' weights across the network. As described in detail by Sundararajan et al.[18], the IG score calculated for a specific sample is generated for each input dimension on each output dimension by accumulating gradients along the path from the input to output. Thus, applying IG to MD-AD allows us to achieve sample-specific gene importance for each neuropathological phenotype predicted by MD-AD. In addition, by treating the last shared layer as the "output" of the MD-AD model (i.e., by temporarily removing all subsequent layers), IG is also able to identify gene-level importances for nodes in MD-AD's last shared layer. As described next, we use IG applied both to the phenotype predictions and last shared layer nodes to interpret MD-AD's learned representations.

*Obtaining IG scores*. We note that for each MD-AD model (of the 100 re-trainings), we apply the IG algorithm for each sample, which generates sample-specific IG scores for genes on each output. Thus, for each MD-AD model, we generate a (# samples × # genes × # output nodes) matrix providing sample-level gene importances for output nodes. Using the standard approach for a single MD-AD model provides sample-specific importances for each gene on each output phenotype. We additionally generate a modified MD-AD network with all layers beyond the last shared layer removed to obtain sample-specific IG scores for genes on all nodes in the last shared layer. Thus, for each of the 100 MD-AD models, we have a (# samples × # genes × # output phenotypes) matrix of gene importance for neuropathology predictions, as well as a (# samples × # genes × # last shared layer nodes) matrix of gene attributions for the last shared layer of the network. As described in the following section, we derive insights from the consensus MD-AD model by aggregating these IG values in various ways.

*Aggregating gene importance scores for nodes*. For both the output nodes (six neuropathological phenotype predictions) and the last shared layer nodes, we have (# samples × # genes × # nodes) IG matrices for each MD-AD run as described above. Now, we describe how we are able to aggregate across samples (and ultimately runs) to obtain a final gene ranking for each (output or last shared layer) node. First, for each MD-AD run, we generate a gene ranking for each of these nodes using a weighted average. Our weighted average uses the following weights: +1 for samples from individuals with high Braak and CERAD scores, −1 for samples from individuals with low Braak and CERAD scores, and 0 otherwise. Thus, the genes with the highest aggregated IG scores are those for which high IG scores coincide with high node outputs. This approach is used for both ranking genes' relevance to neuropathology in the MD-AD framework, and for annotating the last shared layer nodes, as described in the next sections.

**Constructing and annotating MD-AD consensus nodes**. Because deep neural networks have non-convex loss functions, randomness in our training procedure produces networks with different weights from run to run. In order to capture robust nodes and highly relevant genes, we repeat our training procedure 100 times, in order to simulate a "consensus network". As shown in Supplementary Figure 8a, we construct "MD-AD consensus nodes" by clustering nodes from many

runs: (1) we train 100 MD-AD networks, (2) we obtain the last shared layer node outputs for all samples and normalize them (0-mean, unit variance), (3) we combine all nodes across all runs and then cluster them using k-means (where the dimensions used to calculate similarity are samples) with $k = 50$, (4) we summarize each cluster of nodes by their medoid. Thus, for each sample, the MD-AD consensus embedding is made up of 50 nodes, which are medoids of clusters generated from 100 re-trainings.

In Supplementary Figure 6b, we provide a visual overview of the MD-AD consensus embedding generated as described above. To provide a simple view of clusters, we select a subset of samples for which we have clear high or low pathology, excluding ambiguous cases. We include (1) individuals with Braak stage of at least five and CERAD scores at least three (i.e., "moderate"), or (2) individuals with Braak stage of 3 or lower and a CERAD score of 1 (i.e., "absent") who are at least 85 years old and have no dementia. Case 1 captures all individuals with pathologic AD diagnoses (with and without dementia), whereas case 2 captures all individuals considered "resistant" to AD due to their old age but lack of cognitive or neurological decline (consistent with previous literature, e.g., Latimer et al.[53]). To annotate each node in the consensus embedding, we display their correlations with various phenotypes and covariates, as well as their enrichment for REACTOME pathways.

*Correlations.* For each variable (neuropathological phenotypes, high-level AD phenotypes, and covariates), we compute the correlation $-\log10(p$ value) between the variable and each consensus node output. In Supplementary Figure 6c, a high $-\log10(p$ value) indicates that a node captures (or is highly linearly related to) a variable.

*Pathway enrichment.* Beyond relationships between nodes and various phenotypes, we annotated nodes with which gene sets are relevant to their outputs. First, in order to identify relevant genes to each consensus node, we use IG scores. As described in the "Model interpretation with Integrated Gradients" section above, for each run, we aggregate IG scores across samples to obtain a weighted average of gene importance scores for each last shared layer node. Because our consensus last shared layer nodes are actually individual nodes sourced from various runs of MD-AD, we simply combine the aggregated IG scores from the relevant nodes across these runs. For each MD-AD consensus node, this method, therefore, provides us with a ranking over all genes by their importance. We then test for enrichment of REACTOME pathways[54] in these gene rankings via GSEA[30,31] to identify whether certain pathways seem to be involved in the activation of these nodes. Enriched pathways for the MD-AD consensus nodes are shown in Supplementary Figure 6d. Supplementary Data 3 provides detailed annotations for each node.

**Identifying MD-AD's top genes.** In order to identify genes that drive MD-AD predictions, we used IG[18] to provide importance estimates of each gene on the predicted outcomes. In order to improve model stability, we calculate gene rankings based on 100 re-trainings. As described in the "Model interpretation with Integrated Gradients" section above, after each run of training, we take our trained model and apply IG for each sample to get the importance of each gene on each neuropathological phenotype prediction. We next aggregate our IG scores into gene rankings by calculating the ranks of each gene (for each phenotype) in each run and then averaging across runs to obtain consensus gene ranks. For each phenotype (see Supplementary Figure 8 for illustration). Thus, the gene with the highest consensus IG score (i.e., score close to 1) is the gene with the highest average rank across runs (most positively associated with the neuropathological phenotype), and the gene with the lowest consensus IG score (i.e., score close to 0) is the gene with the lowest average rank across runs (most negatively associated with neuropathology). Although we generate these consensus rankings separately for each phenotype, we again average across the six phenotypes to obtain our final MD-AD consensus IG scores. We note that 100 re-trainings are more than enough to converge to a stable gene ranking (Supplementary Figure 8c). The top genes for MD-AD are shown in Fig. 4a, and enriched REACTOME pathways in the top-ranked MD-AD genes (via GSEA) are shown in Fig. 4b. The full gene ranking, generated separately for each neuropathological phenotype, is provided in Supplementary Data 4.

For comparison with a linear gene ranking method, we also generate correlation-based gene rankings as follows: we calculate the correlation coefficients between each gene's expression level and each neuropathological phenotype (across all samples in our dataset), and then percentile rank the genes by their average correlation coefficients across all six phenotypes (with 0 for the most negatively correlated and 1 for the most positively correlated gene with high pathology). Our final correlation-based gene ranking is the average over the phenotype-specific rankings. Comparisons between REACTOME categories represented in the top MD-AD vs correlation-based rankings are shown in Fig. 4c.

**Calculating non-linear effects for MD-AD genes.** As a deep learning method, MD-AD has the capacity to identify non-linear relationships among genes' expression levels and neuropathological phenotypes. These non-linear relationships may reveal an implicit capture of interaction effects with other covariates observable from expression data. Thus, we sought to investigate the presence of interactions between sample-level covariates and specific genes in their contributions to the MD-AD predictions.

*Generating sample-level gene importances scores.* To simplify our analyses, we generate consensus IG scores for each gene within each sample as follows: for each sample and gene, we average over the gene's IG weights across both neuropathological phenotypes and runs in order to obtain its average importance for general neuropathology across all runs.

*Measuring interaction effects.* To monitor the presence of interaction effects in gene importance scores, we modeled the consensus per-sample IG scores as a linear combination of a gene's expression level, a covariate of interest, and the interaction of the two. Specifically, $score_{g,i} = a\,expr_{g,i} + b\,feat_i + c\,expr_{g,i}feat_i + d$, where $score_{g,i}$ is the consensus IG value for gene $g$ and sample $i$, $expr_{g,i}$ is the sample $i$'s expression level for gene $g$, and $feat_i$ is sample $i$'s value for the covariate. Based on this representation, we consider there to be an interaction effect between a gene and feature on its importance in the MD-AD model if the learned $c$ coefficient is statistically significant ($p < 0.05$, after FDR correction over all genes). We primarily focus on identifying interaction effects with sex ($feat_i = 1$ if sample $i$ comes from a male, 0 otherwise), and rank interactions between genes and sex for MD-AD based on the $-\log10(p$ value) of the interaction term.

*Gene set enrichment.* We evaluated whether sex-differential genes were enriched for the following gene sets: (1) REACTOME pathways[54] and (2) microglial cluster gene signatures from a recent single-cell RNA-seq analysis of microglial cells from autopsied aging brains[38]. To evaluate whether the list of sex-differential MD-AD genes is enriched for gene sets of interest, we use Fisher's exact tests to evaluate the significance of the overlap between all sex-differential genes and members of each gene set. Next, to evaluate whether the top MD-AD sex-differential genes are enriched for the same gene sets, we perform Fisher's exact tests again, but this time only consider the top 100 MD-AD genes in the calculations.

**External validation: blood gene expression.** To evaluate the ability of MD-AD to transfer to blood gene expression data, we downloaded publically available AddNeuroMed cohort data from GEO (GSE63060 and GSE63061, which we refer to as Blood1 and Blood2, respectively). Details about the AddNeuroMed samples are provided in Supplementary Data 5. As with the other validation datasets, each blood dataset was normalized such that each gene's expression values have the same mean and variance as the processed MD-AD expression data. Because each blood dataset had a different set of available genes, for each dataset, we re-trained MD-AD consensus models for brain samples with only the genes available between them and blood samples (12,104 and 11,392 genes for Blood1 and Blood2, respectively). Because these blood samples came from living participants, we do not have access to the many neuropathology variables available across the brain samples. Instead, we assess whether MD-AD's predictions align with individuals' cognitive diagnosis of CTL, MCI, or dementia.

We evaluate the effectiveness of the MD-AD model by comparing predicted MD-AD pathology scores between CTL and MCI individuals, and between CTL individuals and individuals with dementia via two-sided $t$ tests (together, and split by age). To evaluate the MD-AD embedding for blood samples, separately for each blood dataset, we obtain the last shared layer embeddings of both the MD-AD brain expression samples and blood samples from the first round of training.

**Reporting summary.** Further information on research design is available in the Nature Research Reporting Summary linked to this article.

## Data availability

No new data are generated in this study. All datasets used were either publically available or available subject to data-use terms and conditions as described below. Most human brain gene expression and phenotype datasets were obtained via the AD Knowledge Portal Synapse platform (doi: 10.7303/syn2580853). Access to these datasets may only be obtained after registering for a Synapse.org account, agreeing to acknowledge data used in any publications, and submitting a data-use certificate (separately as needed for each dataset). Our study uses the following datasets (with listed Synapse IDs; URLs): ACT (syn5759376), ROSMAP (syn3219045; https://doi.org/10.1038/s41593-018-0154-9), MSBB (RNA Sequencing: syn3159438, Microarray: syn3157699), Mayo Clinic Brain Bank (syn5550404; https://doi.org/10.1038/sdata.2016.89). All other human brain, mouse brain, and human blood datasets were downloaded from the Gene Expression Omnibus (GEO). The following datasets are publically available for download (with listed accession codes; URLs): HBTRC (GSE44772; https://doi.org/10.1016/j.cell.2013.03.030), human blood gene expression and phenotype data from the AddNeuroMed cohort (GSE63060; and GSE63061), Mouse brain gene expression samples, and associated phenotypes (GSE64398). Our study reports pathway enrichment for our results with respect to publically available gene sets. These include REACTOME and KEGG pathways available from MSigDB (c2 pathways v7.0; http://www.gsea-msigdb.org/gsea/msigdb/genesets.jsp?collection=C2). We also compared our results with gene signatures from Olah et al[38]. (Supplementary Data 5 in their publication) and Mathys et al[8]. (Supplementary Table 6 in their publication). Source data are provided with this paper.

## Code availability

All code for our study, including code to train the MD-AD model and to generate all figures included in the manuscript, are available at https://github.com/suinleelab/MD-AD (archived at https://doi.org/10.5281/zenodo.5043447).

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

## Acknowledgements

This work was funded by the National Institutes of Health (R35 GM 128638, R01 AG061132, R01 AG036836, U01 AG046152, and U01 AG061356) and the National Science Foundation (DBI-1552309). We thank the participants of the studies who have generously donated their

time and brains to these projects. We also thank Sandy Kaplan for her help in editing our manuscript. The results published here are in part based on data obtained from the AMP-AD Knowledge Portal (https://doi.org/10.7303/syn2580853). ACT study data were generated from postmortem brain tissue collected through the Adult Changes in Thought study (E. Larsen), and The University of Washington ADRC (T. Montine) (U01AG046161; P50 AG025688; P30 NS055077). MSBB study data were generated from postmortem brain tissue collected through the Mount Sinai VA Medical Center Brain Bank and were provided by Dr. Eric Schadt from Mount Sinai School of Medicine (U01AG046170, U01AG046161, RF1AG057440, R01AG050986). ROSMAP Study data were provided by the Rush Alzheimer's Disease Center, Rush University Medical Center, Chicago. Data collection was supported through funding by NIA grants P30AG10161, R01AG15819, R01AG17917, R01AG36836, U01AG46152, U01AG46161, the Illinois Department of Public Health, and the Translational Genomics Research Institute (genomic). Additional phenotypic data can be requested at www.radc.rush.edu. MayoRNASeq study data were provided by the following sources: The Mayo Clinic Alzheimers Disease Genetic Studies, led by Dr. Nilufer Taner and Dr. Steven G. Younkin, Mayo Clinic, Jacksonville, FL using samples from the Mayo Clinic Study of Aging, the Mayo Clinic Alzheimer's Disease Research Center, and the Mayo Clinic Brain Bank. Data collection was supported through funding by NIA grants P50 AG016574, R01 AG032990, U01 AG046139, R01 AG018023, U01 AG006576, U01 AG006786, R01 AG025711, R01 AG017216, R01 AG003949, NINDS grant R01 NS080820, CurePSP Foundation, and support from Mayo Foundation. Study data includes samples collected through the Sun Health Research Institute Brain and Body Donation Program of Sun City, Arizona. The Brain and Body Donation Program is supported by the National Institute of Neurological Disorders and Stroke (U24 NS072026 National Brain and Tissue Resource for Parkinsons Disease and Related Disorders), the National Institute on Aging (P30 AG19610 Arizona Alzheimers Disease Core Center), the Arizona Department of Health Services (contract 211002, Arizona Alzheimers Research Center), the Arizona Biomedical Research Commission (contracts 4001, 0011, 05-901 and 1001 to the Arizona Parkinson's Disease Consortium) and the Michael J. Fox Foundation for Parkinsons Research. HBTRC study data were generated from post-mortem brain tissue collected through the Harvard Brain Tissue Resource Center and were provided by Dr. Eric Schadt from Mount Sinai School of Medicine (U01AG046170).

## Author contributions

N.B.W., S.C., S.M., and S.-I.L. designed the study. N.B.W., E.W., and P.S. carried out the analyses. N.B.W., P.L.D.J., S.M., and S.-I.L. wrote and edited the manuscript. S.C., S.M., and S.-I.L. supervised the study.

## Competing interests

The authors declare no competing interests.
