## [Peer Review File · Nature Communications]

Reviewers' Comments:

Reviewer #1:

Remarks to the Author:

General: This is a highly interesting article, which demonstrates the potential of modern neural network based approaches in Alzheimer's Disease research and in particular for gene expression analysis. The article is very well written and contains a careful analysis of the proposed model. Nonetheless there are a couple of points that should be addressed prior to publication.

Major:

1. All clinical studies are unavoidably biased by patient selection criteria. The biggest concern when pooling together data from different studies is thus that a model trained over joint data is biased towards the most abundant sub-cohort in the training data. The authors should thus initially show and statistically compare clinical characteristics (age, gender, APOE4 status, cognitive and neuropsychological assessment scores) of patients across studies.
2. Neuropathological measures might differ in their statistical distribution across studies. The authors should thus show a range normalized version of the RMSE.
3. The authors mention 1,758 gene expression profiles from 925 distinct individuals, suggesting that there are individuals with more than one gene expression profiles. How was this situation handled in the training and testing of models? Note that data from the same individual can be expected to be more correlated than data from different individuals.
4. It is overall not entirely clear, where the authors see the utility of their model. Do they think MD-AD is good for improving the biological understanding of AD or just for demonstrating the technical possibility to integrate gene expression data from different studies? Note that the model has little clinical utility.
5. The authors compared their rather complex MD-AD model against a linear one. Which predictor variables did this model contain? The same as MD-AD? What was the computation time for training MD-AD?
6. In the supplements the authors write that MD-AD was actually trained with the first 500 principal components extracted from the original gene expression data. Note that PCA is a linear projection method, while MD-AD is non-linear. If the authors believe that non-linear projections (as internally done via a neural network) offer a benefit compared to a linear approach, then this is actually an inner contradiction. Moreover, the number of 500 PCs seems rather arbitrary. Which fraction of total variance is explained by them? I think that overall a much more rigorous and consequent approach would have been to include a strong dimensionality reduction and regularization as part of MD-AD.
7. It is not clear, how MD-AD the was regularized and tuned. For example, were drop-out units used? Which activation functions were used? How were hyper-parameters tuned? Please provide a detailed list of hyper-parameters of the model in the Supplements. Please also indicate whether the choice of the architecture among those shown in Figure S2 was actually part of the hyper-parameter tuning (and thus done WITHIN the cross-validation loop) or not. In the latter case reported prediction performances would be over-optimistic.

Reviewer #2:

Remarks to the Author:

In this work, Wang et al introduce a new analysis framework (MD-AD) that aims to assess non-linear relationships between gene-expression levels and (disease) phenotypes. The framework relies on a single neural network modeling approach that jointly models information from multiple cohorts. The analysis framework is applied to a large collection (N=1758 samples across 9 brain regions) of post-mortem transcriptomic data obtained from Alzheimer's Dementia (AD) patients and controls. Predictions are further confirmed in blood and across species using AD-relevant mouse models. Results show that the MD-AD algorithm is predictive of neuropathology related to AD and may highlight relevant biological pathways by indicating the genes mainly driving the prediction.

Although I believe this method is of potential interest as it integrates information from multiple cohorts and phenotypes and seems to have predictive ability, I do think that with the information

currently provided, it is difficult to assess whether the proposed model is unbiased. Assumptions of the algorithm are not mentioned, and several crucial decisions are not explained in detail. This can be improved. Also, with regard to insight into biological pathways, it is unclear to what extent the current method provides novel insight above e.g. current genome-wide association studies.

I have listed several points below where I think clarification is needed:

- The proposed analysis framework can accommodate sparsely labeled data (i.e. measures that are not available across all cohorts), but in order to do so several assumptions have to be made. These assumptions are however not explicitly discussed, and therefore it is currently not possible to assess how realistic these are, under what circumstances they may be unlikely to hold, and how this might influence results. Please add a discussion on the (implicit) assumptions of the MD-AD framework.
- It is stated that the framework can deal with partly overlapping phenotypes across cohorts, but can the analysis framework also accommodate cohorts that have non-overlapping phenotypes (but maybe correlated phenotypes)?
- p2 lines 63-64: ‘..which tend to capture...’ Please make this statement more concrete, as current methods do allow to correct for the presence of different cohort by e.g. including cohort as a covariate
- p 3 lines 75-77 Predicting...time: This sentence can be made more explicit. Also, it seems phenotypes are used both to denote the end-phenotype (i.e. the case control status), clinical features, and expression levels. Please check throughout the manuscript and make this clearer where needed. For example, in line 78 it is not directly clear whether ‘joint training’ refers to multiple different diagnoses or multiple different biomarkers of AD
- p3 line 79: Here the authors mention ‘increased sample size’, but what is really meant, it seems, is the ability to analyze multiple cohorts at the same time, instead of each cohort separately. This does lead to increased sample size, but that does not seem to be the main point here. So maybe rephrase this sentence. If indeed increased sample size is meant here, it requires a quantification of the comparison
- p3 line 81: The reason why multi-layer perceptrons can be of use should be briefly explained here, as the current statement is unjustified
- p3 line 84: What does ‘these advantages’ refer to? I assume it refers to training the MLPs jointly, but there are no specific advantages mentioned in the previous 2 sentences, so please adjust
- The statement in lines 86-88 does not seem appropriate in the introduction, maybe move that to the discussion/conclusion section or re-phrase (e.g. ‘MD-AD is designed to ...’)
- How are correlations between gene expression levels taken into account?
- How were different post-mortem delays between samples (and most importantly- were these comparable between cases and controls) taken into account? (as these may possibly influence gene expression levels in a non-systematic way)
- ACT, MSBB, and ROSMAP cohorts were used for training, but which cohorts were used for predictions discussed on p. 4-5? From the supplementary materials (p3) it seems that each dataset was split into 5 training and test sets, but that means the training and test sets are only semi-independent. Please explicitly state this (on p4 or 5). Also, it is important to mention that there was no sample overlap in these cohorts, this can be added in the main text. Only on p5 the ultimate out of sample prediction is mentioned, but prediction should always be out of sample to avoid bias, so please discuss why in-sample prediction (i.e. prediction in the sample that is also used for training) is chosen as well, as this is generally not considered good scientific practice
- p4 line 116 and p6 lines 168-169 the sample sizes are mentioned, but this is not split into cases and controls. Please provide numbers for cases and controls separately
- p5 lines 163-164 states ‘we observed that adding new samples improved performance for a phenotype even when the phenotype in question was not measured in the new samples’ The authors explain this by saying that the shared representation learned by the algorithm captures the underlying biological signal - but how? And is this not dependent on the correlation between the phenotypes (and thus not a general effect)?
- An implicit assumption seems to be that gene-expression patterns are causally related to AD pathology. However, the reverse may also be true: that due to the forming of plaques or the use of medication, DNA methylation patterns are altered, and differences in gene expression are a consequence rather than a cause of the disease, which is difficult to resolve using post-mortem

transcriptomic data, although the mouse data can shed some light on this. This can be briefly discussed.

- p8 line 246: Briefly add what the Integrated Gradient algorithm entails and why it is appropriate
- lines 260-263: Gene importance scores tended to yield more specific pathway enrichment than scores based on correlations - can the authors state whether they think 'more specific' means 'closer to the truth' (as it seems to suggest now) and if so why? This statement needs an interpretation by the authors here
- lines 260-263: How was the 'gene correlational score' calculated? Please explain
- lines 272-273: 'Of the ... correction' Here it is important to know whether this analysis was based on IG scores that were calculated in the total sample, or separately in males and females, thus capturing a fixed effect of sex or looking at less systematic differences (the latter). Also, were expression profiles of genes on the X-chromosome included?
- Table S4: I am bit surprised to see that the APOE gene has such a low MD-AD consensus score, while it is one of the strongest genetic effects on AD - how would the authors explain this?
- Using multiple data-sets was there some sort of data processing built into a pipeline to ensure data integrity? (Was there a re-scaling of the data done? What happened to outliers?)

Typos:

- p3 suppl. materials: hyperparamters should be hyperparameters
- line 91 (figure) 1d should be (figure 1d)

Thank you for considering our manuscript “Unified AI framework to uncover deep interrelationships between gene expression and Alzheimer’s disease neuropathologies” for review by *Nature Communications*, and for allowing us to submit a point-by-point response and incorporate the comments that we have received into a new revision of the manuscript.

Decision on 1/25/2021

Thank you again for submitting your manuscript "Unified AI framework to uncover deep interrelationships between gene expression and Alzheimer’s disease neuropathologies" to Nature Communications. We have now received reports from 2 reviewers and, on the basis of their comments, we have decided to invite a revision of your work for further consideration in our journal. Your revision should address all the points raised by our reviewers (see their reports below).

When resubmitting, you must provide a point-by-point response to the reviewers’ comments. Please show all changes in the manuscript text file with track changes or colour highlighting. If you are unable to address specific reviewer requests or find any points invalid, please explain why in the point-by-point response.

Important: *In addition to the above, you must comply with the following editorial requests; we will not be able to proceed with your revised manuscript otherwise. Please also see the Nature Communications formatting instructions, which you may find useful while preparing your revised manuscript.*

We would also like to thank the reviewers for their careful consideration of this manuscript and many suggestions for improvement. In response to the reviewers’ comments we have made major changes that we feel substantially improve the manuscript and address the reviewers’ concerns.

The two major changes include:

- 1. Adding clarifying details about our methods.** Based on questions and concerns from both reviewers, we have added information about the data and preprocessing steps and additional details about our model training and evaluation. We have also added additional contextual information such as modeling assumptions and details of the integrated gradients algorithm.
- 2. New cross-validation experiments.** Both reviewers raised valid concerns about potential bias related to our approach to cross-validation experiments. First, per Reviewer 1’s comments, we generated new cross-validation splits *by individual* rather than by sample. Second, per Reviewer 2’s comments, we used an additional preprocessing step to *control for measurement-related covariates* in our gene expression data before model training and evaluation. We have conducted these experiments and included the results as supporting material. As we described, the results were consistent with our initial cross-validation experiments.

Below is a point-by-point response to the referee comments. The reviewer's comments are in blue and italics, and our replies are in black. We have additionally included some excerpts from our revised manuscript in gray boxes for your convenience.

Reviewer #1 (Remarks to the Author):

General: This is a highly interesting article, which demonstrates the potential of modern neural network based approaches in Alzheimer's Disease research and in particular for gene expression analysis. The article is very well written and contains a careful analysis of the proposed model. Nonetheless there are a couple of points that should be addressed prior to publication.

We thank the reviewer for the positive feedback and appreciate their careful consideration of our analyses. Their concerns are addressed below.

Major:

1. All clinical studies are unavoidably biased by patient selection criteria. The biggest concern when pooling together data from different studies is thus that a model trained over joint data is biased towards the most abundant sub-cohort in the training data. The authors should thus initially show and statistically compare clinical characteristics (age, gender, APOE4 status, cognitive and neuropsychological assessment scores) of patients across studies.

We agree that an uncareful aggregation of data indeed may not reveal any new insights besides what could be gleaned from the “majority” data type/dataset. We have added **Supplementary Table 2** (referenced on line 129) which provides an overview of clinical characteristics, gene expression-related covariates, and neuropathological phenotypes for each cohort. We have shown with several careful experiments (**Figures 2, 3 and 7**) that joint training allowed MD-AD to learn generalizable information that led to improved performance both within these cohorts **and** on multiple external validation datasets. These results strongly suggest that our model has not merely learned (or possibly overfitted to) the particular characteristics of one study.

In particular, we would like to highlight a relevant result included in our paper. In **Figure 2b** (and described on lines 156-175) we assessed the MD-AD's performance when trained on different subsets of the three datasets, and in a setting where one complete cohort was left out in some of the training experiments. When the entire ROSMAP study was left out of training experiments, we found that training with additional datasets tended to improve *ROSMAP test samples'* (despite differences in their clinical characteristics). We displayed the results specifically for the ROSMAP dataset because it was the only dataset with labels for all six phenotypes; however MSBB was the most abundant sub-cohort (879 samples vs. ROSMAP's 542). Although one might expect that adding MSBB samples to the training may lead the model to bias the learned model towards MSBB's distribution and become less relevant to ROSMAP samples, we instead see that for almost all phenotypes, training on ROSMAP *and* MSBB training data improves test prediction performance compared with training on ROSMAP samples alone (and we see further improvements when adding ACT samples) (**Figure 2b**). We believe this reflects

MD-AD's improved ability to learn from the increased samples by adding cohorts; this allows MD-AD to learn some true underlying biological signals beyond any differences in covariates or batch effects.

2. Neuropathological measures might differ in their statistical distribution across studies. The authors should thus show a range normalized version of the RMSE.

We thank the reviewer for the suggestion. In order to ensure that neuropathology variables were on similar scales across studies, within each study, we first normalized each phenotype's values to range from 0 to 1. Because various phenotypes had different distributions, our reported test set MSEs are normalized such that we divide each MSE by the variance of the phenotype label in the test set. We neglected to add these details in our original manuscript, but the term for our approach is $1-R^2_{CV}$ (Quan, 1988). We have added this detail to the revised manuscript on line 141, and have additionally updated axis labels and captions in **Figures 2a** and **2b**, **Supplementary Figures 2a**, **11a** and **11b** accordingly. We would also like to note that while the use of $1-R^2_{CV}$ (test MSE divided by the labels' variance) allowed us to more easily compare performance *between phenotypes* versus an unnormalized MSE, such a choice does not affect the comparisons we've made *across methods* (e.g., MD-AD vs. MLPs in **Figure 2a**, because methods were compared within the same phenotype).

Quan, N. (1988). The Prediction Sum of Squares as a General Measure for Regression Diagnostics. *Journal of Business & Economic Statistics*, 6(4), 501-504. doi:10.2307/1391469

3. The authors mention 1,758 gene expression profiles from 925 distinct individuals, suggesting that there are individuals with more than one gene expression profiles. How was this situation handled in the training and testing of models? Note that data from the same individual can be expected to be more correlated than data from different individuals.

Thank you for your thoughtful question. MSBB and ACT include multiple brain regions, and hence some individuals are represented multiple times (with data from different brain regions). In our initial experiments, because samples were collected from different brain regions, we treated them as independent. We reasoned that because the trained MD-AD model performs well on many external datasets, including cross tissue and cross species, the risk of overfitting due to non-independence of a few training/test samples is low.

However, to address your valid concern, we repeated our internal validation experiments as follows. We generated five new training/test splits (along with 5-fold cross-validation splits within each training set) as illustrated in **Supplementary Figure 1e**, but this time we *split by individual* rather than directly by samples. We performed the same cross-validation and hyperparameter selection process as was done for our original dataset splits, and our resulting prediction performance and last shared layer evaluations are shown in (newly added) **Supplementary Figure 11c-d**.

Supplementary Figure 11

In these experiments, we find that MD-AD (as well as the baseline methods) provides very similar prediction performance when trained and evaluated on fully separated training and test sets (**Supplementary Figure 11c**), suggesting that our original results did not hinge on the similarity of samples between the training and test set. We similarly find that MD-AD continues to produce embeddings which capture both neuropathology phenotypes and higher-level AD phenotypes more consistently than alternative approaches (**Supplementary Figure 11d**), although these results are not as consistent with the original splits (perhaps because individual intermediate nodes may be noisier than the final predictions of the model).

Furthermore, we note that these experiments include full hyperparameter optimization as well. We found that the hyperparameters selected using the new splits are similar to the original final model selected from our original splits with the exception of a single hyperparameter (kernel regularization of .001 for our original splits, compared with .00001 for the new splits). However, we find that in our analyses with the new splits, a model trained on our originally selected set of hyperparameters has very similar performance to the newly selected set of hyperparameters (see figure to the right).

Together, these results indicate that our choice to split samples randomly produced very similar findings to the alternative of splitting samples pseudorandomly by individual. Thus, to expedite our revision, we have elected to keep the results and selected models from our original dataset splits.

We thank the reviewer for raising this valid concern, and have added our new experiments with alternative training and test splits (by individual) to our **Supplementary Methods** (lines 214-237) and **Supplementary Figure 11c-d**.

4. It is overall not entirely clear, where the authors see the utility of their model. Do they think MD-AD is good for improving the biological understanding of AD or just for demonstrating the technical possibility to integrate gene expression data from different studies? Note that the model has little clinical utility.

We thank the reviewer for their comment. In fact, we argue that the contribution of the MD-AD framework includes both points suggested above: improving biological understanding *and* demonstrating a framework which integrates gene expression data from different studies.

First, we want to clarify that our end goal is not to merely predict pathology (of course, this would not be of clinical benefit, as brain expression samples come from post-mortem brains). Training to *predict* neuropathology provides a yardstick to measure how well we can identify “important” genes and pathways that yield an understanding of disease mechanisms. These led to new biological insights into underlying mechanisms of AD, including microglial activity and sex differences, as described in results and discussions.

Second, our paper provides a demonstration of our general framework of integrating expression datasets from different cohorts in a way that is generalizable between them and even to new datasets, which is applicable to other diseases. We believe this aspect will be particularly impactful as new datasets emerge.

Additionally, even in terms of clinical utility, we argue that although our model was trained with data from postmortem brains and thus is has immediate clinical utility, we believe future extensions of the MD-AD model (perhaps retrained with growing blood-based expression datasets) could in fact be clinically useful. As shown in **Figure 7** and described on lines 318-332, our MD-AD model trained on brain gene expression data successfully distinguishes between blood-based gene expression signatures in individuals with MCI or dementia and healthy controls. This demonstrates an important first step towards accurately identifying AD cases from blood-based expression signatures in living people.

We have added additional clarification about the contributions of our model to the end of the Discussion section (lines 406-410).

5. The authors compared their rather complex MD-AD model against a linear one. Which predictor variables did this model contain? The same as MD-AD? What was the computation time for training MD-AD?

In order to fairly evaluate the baseline models, we built linear models as consistently as possible as with MD-AD. We built *all* models with the first 500 PCs of gene expression data as our input and phenotypes as our output(s) using tensorflow using Keras. Each linear model was simply made up of an input layer (taking 500 PCs) and an output node (one of the phenotypes). To construct the MLP networks, we used the same set-up as with the linear models, but added hidden layers and dropout layers between them (as illustrated in **Supplementary Figure 1a**). Finally, for MD-AD, we used a similar structure as used by the MLPs, but instead of generating six separate models, we combined the networks by using the same first inputs and two two hidden layers which then branch off towards six separate output nodes. Thus, the linear models, MLPs, and MD-AD models share the same predictor variables.

A single MD-AD model (trained on the full dataset) took about 350 seconds to train on a single Nvidia GeForce GTX 980 Ti GPU. Note that our final MD-AD consensus approach involves training the model 100 separate times.

We thank the reviewer for the questions and have added these details to a new sub-section of the Supplementary Methods (lines 134-141). In the main text on line 135, we now direct the reader to the Supplementary Methods for details about model training.

For your convenience, we provide the updated section below (Supplementary Methods lines 134-141):

Construction of Models. For comparison to MD-AD, we generate six analogous MLP networks with un-shared representations, and six linear models containing no hidden layers, to serve as baseline models (see Supplementary Figure 1a). All models were built using Tensorflow and Keras packages, and were constructed as consistently as possible, with the same inputs. The MLP baseline model was identical to the MD-AD model except with only a single branch of task specific layers. Similarly, the linear baseline models were identical to the MLP baselines but with all hidden layers removed. Each model was trained on a single Nvidia GeForce GTX 980 Ti GPU. Although training time may vary across machines, we found that training the MD-AD model on the full dataset took about 350 seconds on average.

6. In the supplements the authors write that MD-AD was actually trained with the first 500 principal components extracted from the original gene expression data. Note that PCA is a linear projection method, while MD-AD is non-linear. If the authors believe that non-linear projections (as internally done via a neural network) offer a benefit compared to a linear approach, then this is actually an inner contradiction. Moreover, the number of 500 PCs seems rather arbitrary. Which fraction of total variance is explained by them? I think that overall a much more rigorous and consequent approach would have been to include a strong dimensionality reduction and regularization as part of MD-AD.

We thank the reviewer for the suggestions. While we understand their concerns, we opted for PCA-transformed data for several reasons.

First, although we assemble a relatively large collection of gene expression (GE) samples, we still have a common machine learning problem of high dimensionality ($\# \text{ genes} \gg \# \text{ samples}$). We found that our model was able to train much more stably and efficiently using PCA-transformed features, even compared with a full-gene model with extensive regularization and dropout. Using all genes would have led to a 22-fold increase in the number of weights that would be updated during training of the neural network (from $\sim 300\text{K}$ to $\sim 7\text{M}$ weights). We note our choice to use PCA-transformed gene expression as inputs to our MD-AD model (and baselines) is consistent with prior studies using deep learning with gene expression data. Using PCA-based dimensionality reductions for gene expression data has been successfully employed in prior studies gene expression studies in the cancer domain (e.g., Basavegowda & Dagneu, 2020; Fakoor et al., 2013; Zhang et al., 2018).

Basavegowda, H. S., & Dagneu, G. (2020). Deep learning approach for microarray cancer data classification. *CAAI Transactions on Intelligence Technology*, 5(1), 22-33.

Fakoor, R., Ladhak, F., Nazi, A., & Huber, M. (2013). Using deep learning to enhance cancer diagnosis and classification. In *Proceedings of the international conference on machine learning* (Vol. 28). ACM, New York, USA.

Zhang, D., Zou, L., Zhou, X., & He, F. (2018). Integrating feature selection and feature extraction methods with deep learning to predict clinical outcome of breast cancer. *IEEE Access*, 6, 28936-28944.

Although we used PCA as a pre-processing step in the MD-AD framework to improve stability, we note that our model can theoretically reconstruct the original gene expression data with “enough” principal components (as quantified in the next two paragraphs). In fact, while PCA is a linear transformation, we can think of it as an additional layer appended to the beginning of our MD-AD network, but with no squashing function applied between this “PCA layer” and the first real layer of our network. In that way, we can actually think of the “PCA layer” and the real “first layer” as a single linear transformation over gene expression data which may be tuned by updating weights in the “first layer”. Thus, *despite using PCA-transformed data, the MD-AD model can learn complex interactions among genes*, and thus we do *not* see its use as contradictory to our overall findings that deep learning provides a richer and more accurate representation of genes’ expression patterns in AD. In particular, we highlight the efficacy of using the first 500 PCs below.

Most variance explained by 500 PCs. First, as shown in the inset Figure R1(right), the first 500 PCs explain over 92% of the variance in the entire gene expression data, so by using PCA-transformed features, we were able to dramatically reduce the number of dimensions in our input space prior to fitting the deep models while losing relatively little information. (We have added this figure to the as **Supplementary Figure 1b**.)

Linear prediction performance with 500 PCs approaches performance with all genes.

However, we do concede that by using only 500 PCs, we may lose some information which could be predictive of AD neuropathology. We performed an additional experiment to investigate this potential issue. Because we would like to investigate whether using 500 PCs, a linear transformation, provides informative-enough features of the data for our AD-related goals, we try predicting average neuropathology scores from different sets of PCs using a linear model, and comparing predictive performance to the full set of genes. We use cross-validation to tune the alpha parameter and, based on the same training/testing splits used in our main analyses, find that by the time we include up to 500 PCs, we have reached similar predictive performance between a model trained on PCs versus the raw gene expression features (see new **Supplementary Figure 1d**). This suggests the linear transformation provided by the first 500 PCs retains features of gene expression data that are almost as linearly predictive of AD neuropathology as the full dataset using all genes.

We believe these analyses and explanations provide sufficient justification of our choice to use 500 PCs as a starting transformation for our input data. To provide clarity to the reader, we have added the variance explained chart along with the experiment above to the paper (new **Supplementary Figure 1c-d**), and described our reasoning for using 500 PC inputs in our Supplementary Methods (lines 117-133), and have additionally cited earlier studies that employ similar approaches.

For your convenience, we provide the updated section below (Supplementary Methods 117-133):

Pre-processing with PCA. In order to have efficient and robust training and to reduce overfitting, we apply a principal component analysis (PCA) transformation to the data and use resulting top 500 principal components – a 500-dimensional representation of our 14,591 gene expression values – as the input to the MD-AD and all baseline models. ***This approach is consistent with the use of PCA for pre-processing in other studies that have employed deep learning in gene expression analyses¹⁶⁻¹⁸.*** Our choice to use 500 PCs is supported by some preliminary analyses of AD-related signals captured by various PCs. First, as shown in Supplementary Figure 1b, the cumulative variance explained by 500 PCs is about 92%, indicating that reducing our input features by a factor of about 30 still retains most of the variation in the data. However, we note that by using only 500 PCs, we may lose some information which may be especially predictive of AD neuropathology. To investigate this potential issue, we sought to predict average neuropathology scores from different sets of PCs using a linear model, and compare predictive performance to the full set of genes. We use cross-validation to tune the alpha parameter and, based on the same training/testing splits used in our main analyses (described below), find that by the time we include up to 500 PCs, we have reached similar predictive performance between a model trained on PCs versus the raw gene expression features (Supplementary Figure 1d). This suggests that the linear transformation provided by the first 500

PCs retains features of gene expression data that are almost as linearly predictive of AD neuropathology as the full dataset using all genes.

7. It is not clear, how MD-AD the was regularized and tuned. For example, were drop-out units used? Which activation functions were used? How were hyper-parameters tuned? Please provide a detailed list of hyper-parameters of the model in the Supplements. Please also indicate whether the choice of the architecture among those shown in Figure S2 was actually part of the hyper-parameter tuning (and thus done WITHIN the cross-validation loop) or not. In the latter case reported prediction performances would be over-optimistic.

We thank the reviewer for bringing this up. We have added detailed information about our process for tuning hyperparameters to **Supplementary Methods** lines 143-150 and 186-192, and listed our selected hyperparameters in **Supplementary Table 6**. We have also added a cross-reference in the main text (line 135) to refer readers to the details in the Supplementary Methods. Specifically, we used ReLU activations and drop-out units (with drop out rates of 0.1) throughout the hidden layers of the network, and additionally performed hyperparameter tuning to refine our choices for kernel regularization, gradient clipping, and learning rate (see excerpts below).

With regard to the alternative architectures we evaluated (shown in **Figure S2** in the revised manuscript), these architectures were evaluated using the same hyperparameters as selected for our final MD-AD model (outside of the CV loop). This was because we had already designed MD-AD to have hidden layers both before and after the split in the network, so that complex patterns could be learned both within the shared representation and in the representation dedicated to each task. We explored alternatives as a sanity check to make sure our architecture was not particularly low-performing compared to some alternatives. Thus, while we recognize that the reported performance of some architectures may have been slightly underestimated based on sub-optimal hyperparameters, we were reassured to see that the general performance of the MD-AD framework was relatively robust to changes in split location and number of layers. This detail has been clarified in the **Supplementary Methods** lines 164-170.

For your convenience, we provide excerpts for the revised Supplementary Methods sections below.

New subsection about hyperparameters evaluated (Supplementary Methods lines 143-150):

Hyperparameters. *After some preliminary experiments with single and multi-task neural networks, we decided to train all networks with ReLU activations and drop-out units (with drop-out rates of 0.1), and trained each model for 200 epochs with batch sizes of 20 using adam optimization. These settings were selected because they led to relatively stable and effective predictions. We tended to see some variation in performance based on kernel regularization, and hyperparameters of the optimization method, so for hyperparameter tuning (described below), we performed grid search over the following*

hyperparameters: kernel regularization parameter ($1e-3$ vs $1e-5$), gradient clip norm (0.1 vs 0.01) for the adam optimizer, and the learning rate ($1e-3$ vs $1e-4$).

New subsection describing how hyperparameters were selected for the final models and referencing the table listing the selected hyperparameters (Supplementary Methods lines 186-192):

Final model selection. *Finally, after our in-depth cross-validation and testing scheme was used to evaluate our methods internally, we constructed “final models” for external validation and model interpretation. First, we selected a single set of hyperparameters for each model by ranking each configuration’s prediction performance for each round, and then choosing the configuration with the highest average rank. The selected hyperparameters for each “final model” is provided in **Supplementary Table 6**. We trained “final models” for MD-AD and baselines by each using a single set of hyperparameters on the full dataset.*

Modified subsection clarifying that alternative architectures were evaluated after hyperparameters were selected (Supplementary Methods lines 164-170).

*For MD-AD, we **additionally** explored several **alternative** options for architectures with different amounts of shared and task-specific layers (Supplementary Figure 2b-c). We selected the final architecture (shown in Figure 1a) because we wanted to have multiple hidden layers in both the shared portion and task-specific portion of the network to allow for non-linear interactions to be learned in both the shared representation and in the task-specific branches. **However, when we evaluated alternatives to this approach (using the same selected hyperparameters for our original MD-AD model), we found that alternatives to this approach tended to perform similarly or worse (Supplementary Figure 2b-c).***

Reviewer #2 (Remarks to the Author):

In this work, Wang et al introduce a new analysis framework (MD-AD) that aims to assess non-linear relationships between gene-expression levels and (disease) phenotypes. The framework relies on a single neural network modeling approach that jointly models information from multiple cohorts. The analysis framework is applied to a large collection (N=1758 samples across 9 brain regions) of post-mortem transcriptomic data obtained from Alzheimer’s Dementia (AD) patients and controls. Predictions are

further confirmed in blood and across species using AD-relevant mouse models. Results show that the MD-AD algorithm is predictive of neuropathology related to AD and may highlight relevant biological pathways by indicating the genes mainly driving the prediction.

Although I believe this method is of potential interest as it integrates information from multiple cohorts and phenotypes and seems to have predictive ability, I do think that with the information currently provided, it is difficult to assess whether the proposed model is unbiased. Assumptions of the algorithm are not mentioned, and several crucial decisions are not explained in detail. This can be improved. Also, with regard to insight into biological pathways, it is unclear to what extent the current method provides novel insight above e.g. current genome-wide association studies.

We thank the reviewer for their detailed feedback on our analyses and appreciate their constructive feedback. Their concerns are addressed below.

I have listed several points below where I think clarification is needed:

- 1. The proposed analysis framework can accommodate sparsely labeled data (i.e. measures that are not available across all cohorts), but in order to do so several assumptions have to be made. These assumptions are however not explicitly discussed, and therefore it is currently not possible to assess how realistic these are, under what circumstances they may be unlikely to hold, and how this might influence results. Please add a discussion on the (implicit) assumptions of the MD-AD framework.*

We thank the reviewer for the suggestion. We have added a new section to our Supplementary Methods (lines 66-92) to describe several assumptions made by models, as well as specific assumptions required for the MD-AD framework. We've additionally added a cross-reference to this new section on line 135 in the main text.

Below, we provide an overview of key assumptions in our study, followed by descriptions of how we evaluated or buffered potential failures of these assumptions.

Common assumptions of any machine learning models (not specific to MD-AD):

1. We assume that a single training of each model is representative of all training instances.
2. We assume that samples are “sufficiently” independently and identically distributed (i.i.d.) such that a model trained on these samples should generalize well to new samples from the population of interest.
3. We assume that the true data distribution is smooth such that samples with very similar gene expression values should display similar neuropathology.

For assumption #1 above, while non-convexity of deep learning models can lead to randomness in trained models, we used cross-validation to select hyperparameters across several splits of our data across all models. Further, our final model is trained 100 times to generate ensemble predictions and interpretations to generate representative results (**Supplementary Figure 6c**). Assumption #2 often does not hold in

practice, and this is especially true for our datasets which contain multiple gene expression samples (from different brain regions) from the same individuals. We find that regardless of how well assumptions #2 and #3 hold, overall results displayed in **Figures 2, 3** and **7** highlight the effectiveness of the MD-AD model and particularly, its ability to learn generalizable representations that are transferable across cohorts, and even tissues and species.

Additional assumptions of MD-AD:

1. Deep learning methods rely on the assumption that the data is generated by a composition of (learnable) features in a hierarchical manner.
2. Multitask modeling frameworks hinge on the assumption that there is shared common information across the predicted tasks.

Both of these above assumptions appear to hold, as we see improved performance for single-task MLPs over linear models (assumption #1) and improved performance of MD-AD over the single-task MLPs (assumption #2).

For your convenience, we provide the new section below (Supplementary Methods lines 66-92):

B. Modeling assumptions

For modeling gene expression and neuropathology data, we make several assumptions.

Common assumptions of machine learning models. *First, regarding stability of our models, we assume that a single training of each model is representative of all training instances. In order to buffer the potential failure of the assumption, we used cross-validation to select hyperparameters across different splits of our data across MD-AD, singly trained MLPs and linear models. Further, our final model is trained 100 times to generate ensemble predictions and interpretations, as described in future sections. Second, we assume that our samples are “sufficiently” independently and identically distributed (i.i.d.) such that a model trained on these samples should generalize well to new samples from the population of interest. Third, we assume that the true data distribution is smooth such that samples with very similar gene expression values should display similar neuropathology.*

Additional assumptions of MD-AD: *Deep learning relies on the assumption that the data is generated by a composition of (learnable) features in a hierarchical manner. This allows neural networks with multiple layers to collapse correlation patterns in the input space to generate intermediate embeddings in a way that is useful for prediction¹⁴. Unlike linear models, our deep learning framework does not assume that there’s a linear relationship between the predictors and outcomes, nor does it require normally distributed predictors, or low multicollinearity. Although deep learning relies on relatively few assumptions, in practice, some of these assumptions do not fully hold. In particular, our samples are certainly not i.i.d., as some samples are derived from the same brain. Thus, external validation is invaluable for evaluating the effectiveness of our framework in new settings with no information leakage. The observation that MD-AD transfers well to separate datasets (and even species and*

tissues) implies that our framework is effective regardless of whether these assumptions were fully upheld. Finally, multitask modeling frameworks hinge on the assumption that there is shared common information across neuropathological phenotypes¹⁵. In combining multiple datasets with different sparsity patterns, we additionally assume that this common representation is consistent across cohorts and is generalizable to new data sets. This assumption appears to hold, as demonstrated by the improved test performance of the multi-task network over singly-trained MLPs, and improved ability to generalize well to external datasets.

2. *It is stated that the framework can deal with partly overlapping phenotypes across cohorts, but can the analysis framework also accommodate cohorts that have non-overlapping phenotypes (but maybe correlated phenotypes)?*

We thank the reviewer for raising this interesting question. We believe that MD-AD's improvement over separate learning is attributable to a shared underlying representation that is relevant to the different phenotypes in the study. Thus, even in a case where there are no overlapping phenotypes across cohorts, as long as there is thought to be a shared gene expression-related mechanism underlying the phenotypes (thus, for correlated phenotypes), we would expect the MD-AD framework to be an effective approach.

3. *p2 lines 63-64: ‘.which tend to capture...’ Please make this statement more concrete, as current methods do allow to correct for the presence of different cohort by e.g. including cohort as a covariate*

We thank the reviewer for the suggestion and for pointing out that methods do exist to correct for the presence of cohorts. We have revised our original statement, “...current analysis methods that focus on linear relationships between variables (e.g., module analysis⁹) which tend to capture broader patterns in gene expression that often correspond to cohort-level variations...” as discussed below.

One of the advantages of our approach is to provide a unified framework for modeling several cohorts at once. While it is technically possible to add a cohort indicator as a covariate, it is not desirable for two main reasons: (1) adding a cohort indicator as an input to the models essentially results in *different* functions relating genes with neuropathology phenotypes *for each cohort* - thus, the representations learned would not be universally relevant across cohorts, and (2) using covariate indicators in our models would eliminate the ability to effectively transfer our model to unseen cohorts (i.e., external validation samples, mouse samples, and blood samples).

We have revised the statement in the following ways on lines 62-66 to highlight the fact that MD-AD is able to learn a *unified* representation that captures more than just cohort-level variation which is often an issue for linear-based models. First, we've specified that the reason linear methods would tend towards capturing cohorts rather than disease signals is because the largest variation in gene expression tends to

correspond to technical variation in their measurements (with an added citation noting that this is a common phenomenon). Second, we added an example of an approach which notes this difficulty and circumvents the issue by separately modeling AD across different brain regions and cohorts. We note that this approach requires generating a separate gene representation for each cohort and brain region. Together, we hope this edit helps to clarify why it is difficult to model multiple cohorts' data with simpler methods, and why we find MD-AD's unified approach so promising.

For your convenience, we provide the revised text below (lines 60-69):

A unified approach has been hindered by: (1) the need for “harmonized” phenotypes consistently measured across datasets, and (2) the limitation of current analysis methods that focus on linear relationships between variables (e.g., module analysis⁹) which capture only broad patterns in gene expression data. These often correspond to cohort-level variations which consequently obscure true disease signals¹². To circumvent this issue, one approach has been to identify modules separately across brain regions and cohorts before performing using a consensus approach to cluster them⁹.

Here, we develop MD-AD (Multi-task Deep learning for Alzheimer's Disease neuropathology), a unified framework for analyzing heterogeneous AD datasets to improve our understanding of an expression basis for AD neuropathology (Figure 1a-d).

4. p 3 lines 75-77 Predicting...time: This sentence can be made more explicit.

We thank the reviewer for the suggestion. By “Predicting multiple phenotypes at once biases shared network layers to capture relevant features of these AD phenotypes at the same time,” we meant that, when using multi-task deep learning, an inherent consequence of the network's structure is that the model is biased towards a representation that is relevant to *all* tasks (in this case, the six AD neuropathological phenotypes) as described in detail by Caruana (1993)).

Caruana, R. A. Multitask Learning: A Knowledge-Based Source of Inductive Bias. *Mach. Learn. Proc. 1993* 41–48 (1993). doi:10.1016/b978-1-55860-307-3.50012-5

We see now that by including the specific example of our application, it was not clear that this bias is a general phenomenon in multi-task learning. We appreciate your feedback! We've attempted to make this more explicit by first stating this general feature of multitask learning (with the added citation). This sets us up for the next sentence which explains why we think this is a particularly important feature for our specific goal of modeling AD-related data.

For your convenience, we provide the revised text below (lines 77-82):

*Predicting multiple **outcome variables** at once biases shared network layers to capture relevant features of **all those outcome variables (here, neuropathological phenotypes)** at the same time¹³. This is of critical importance **in our application**: each **neuropathological** phenotype represents a different noisy measurement of the same underlying true biological process, and, as we demonstrate, joint training **with these phenotypes** allows MD-AD to average out the noise to extract the true hidden signal.*

5. *Also, it seems phenotypes are used both to denote the end-phenotype (i.e. the case control status), clinical features, and expression levels. Please check throughout the manuscript and make this clearer where needed. For example, in line 78 it is not directly clear whether 'joint training' refers to multiple different diagnoses or multiple different biomarkers of AD*

We thank the reviewer for raising this issue and have attempted to resolve any confusion in the text. In our revised manuscript, we only use the word “phenotype” in the following contexts:

- In the introduction, we generally use the term “phenotype” broadly to refer to any characteristic or outcome (e.g., line 52), and have left those references as-is.
- In the rest of the paper, we specifically use the term “neuropathological phenotype” to refer to the measurements we directly predict in the model (e.g., lines 79, 128, 171, 182).
- When describing higher level AD characteristics (not used during model training, but used to evaluate the model’s generalizability), we refer to them as “higher-level AD phenotypes” (e.g., line 234).

Specifically, for line 78, we have updated the sentence as described above to more clearly indicate that neuropathological phenotypes are the outcome variables used during training, as shown below:

*Predicting multiple **outcome variables** at once biases shared network layers to capture relevant features of **all those outcome variables (here, neuropathological phenotypes)** at the same time¹³.*

We thank the reviewer for bringing this lack of clarity to our attention, and hope the updated text throughout the manuscript is now more clear.

6. *p3 line 79: Here the authors mention 'increased sample size', but what is really meant, it seems, is the ability to analyze multiple cohorts at the same time, instead of each cohort separately. This does lead to increased sample size, but that does not seem to be the main point here. So maybe rephrase this sentence. If indeed increased sample size is meant here, it requires a quantification of the comparison*

We thank the reviewer for raising this point. Our main goal for this sentence was to point out that, while multi-task learning allows us to analyze multiple cohorts at the same time (which is already a great advantage), the ability to combine cohorts also allows the model to make use of many more samples than

are available from a single cohort. This has the added advantage of making more complex models feasible to train, since most transcriptomic aging brain datasets do not have enough samples to robustly train complex models such as deep neural networks (which we start discussing in the next sentence). In the case of this paper, we now have double the samples that were available from our largest source study (~1800 vs ~900), but this advantage will only grow with the growing number of available datasets. In the Results section, we illustrate this point in **Figure 2b** where we show improvements in ROSMAP test samples' predictions when allowing the model to train on additional datasets (beyond ROSMAP's training samples alone). We see that rather than suffering covariate shifts, we instead see that MD-AD is able to make performance gains from new samples.

We thank the reviewer for pointing out that our sentence did not clearly highlight the benefits of combining cohorts. We have edited this sentence to elaborate on this point and added a quantification of the comparison (lines 82-85), and for your convenience show the edited text below:

Additionally, the increased sample size from combining cohorts (in our case, doubling the number of samples from any individual study) facilitates using deep learning models, which are expressive and able to capture complex non-linear interactions among features¹⁴.

7. p3 line 81: *The reason why multi-layer perceptrons can be of use should be briefly explained here, as the current statement is unjustified*

We thank the reviewer for the suggestion. Because multi-layer perceptrons can capture complex interactions among input genes, they may more accurately capture the complex relationships that exist in genomics studies than linear models do. This is why we originally claimed that “MLPs offer another powerful approach for directly capturing complex relations between gene expression and a phenotype.” We have added a brief explanation to the revised manuscript, as well as references highlighting the use of MLPs with gene expression for other diseases. For your convenience, we provide the revised text below (lines 82-86):

Additionally, the increased sample size from combining cohorts (in our case, doubling the number of samples from any individual study) facilitates using deep learning models, which provide flexibility and the ability to capture complex non-linear interactions among features¹⁴. In particular, multi-layer perceptrons (MLPs) have been used to effectively perform disease classification and prediction from gene expression data¹⁵⁻¹⁷

8. p3 line 84: *What does ‘these advantages’ refer to? I assume it refers to training the MLPs jointly, but there are no specific advantages mentioned in the previous 2 sentences, so please adjust*

We thank the reviewer for the clarifying question. Indeed, “these advantages” was meant to refer to an earlier sentence “This unified framework has key advantages over separately trained models...”, but was too distant from the earlier statement. We’ve adjusted the sentence to clarify that we’re discussing the advantages of MD-AD’s joint training approach.

The revised text is provided below for your convenience (line 89):

We demonstrate that MD-AD’s joint training approach improves prediction accuracy, enabling its predictions to generalize across species and tissue types (Figure 1b).

9. *The statement in lines 86-88 does not seem appropriate in the introduction, maybe move that to the discussion/conclusion section or re-phrase (e.g. ‘MD-AD is designed to ...’)*

We thank the reviewer for the suggestion. We agree that such a statement is better suited for the Discussion section. While revising, we found that the Discussion already contains similar statements (e.g., lines 355-356: “By exploiting the synergy between deep learning and a multi-cohort, multi-task setting, we demonstrated that MD-AD can capture complex, non-linear feature representations that are not learned using conventional expression data analysis methods.”). Thus, we have removed the original sentence from the Introduction. For your convenience, we highlight the removed text below (line 91):

~~*MD-AD’s ability to capture complex non-linear relationships provides an opportunity to gain new insights into the expression basis of AD neuropathology, which were not identified by previous approaches. However, An obvious drawback of deep neural networks is their black-box nature, making it difficult to biologically interpret gene-phenotype associations.*~~

10. *How are correlations between gene expression levels taken into account?*

We thank the reviewer for raising this important question. By composing layers of functions, our neural network models are able to learn flexible representations from input data. The MD-AD model is able to collapse correlation patterns present in the gene expression data at intermediate layers, which is an advantage of deep learning over linear methods that rely on additional assumptions (e.g., uncorrelated features) (LeCun, Bengio, Hinton, 2015). We note that the intermediate embeddings in our models can be seen as different ways of collapsing the correlation patterns among gene expression data in a way that is useful for prediction (as shown in **Figure 3** and **Supplementary Figures 4-5**). Thus, correlations among gene expression levels are implicitly accounted for by MD-AD during model training.

LeCun, Y., Bengio, Y., & Hinton, G. (2015). Deep learning. *Nature*, 521(7553), 436-444.

11. How were different post-mortem delays between samples (and most importantly- were these comparable between cases and controls) taken into account? (as these may possibly influence gene expression levels in a non-systematic way)

We thank the reviewer for raising this concern. We agree that covariates may influence gene expressions in a systematic way, and thus should be critically considered. In our response below, we describe detailed analyses which conclude that differences in post-mortem intervals (PMI) do not appear to be a major factor in our results, as follows: First, we do see a small but significant relationship between neuropathology scores and PMI (as well as RNA integrity number, another potential measurement-related confounder). Second, we explain why we did not correct for them and highlight post-training evidence that these covariates were not highly relevant to MD-AD. Finally, we describe a separate analysis (which has been added to our supplementary methods) which involves re-training MD-AD using covariate-corrected data to show that including these covariates does not seem to change our final results.

First, as the reviewer predicted, there does indeed seem to be a small but significant correlation between neuropathology scores and both postmortem interval (PMI; $r = -0.16$, $p=1e-9$) and RNA integrity number (RIN; $r = -0.09$, $p=0.002$) (Note that we computed correlations between the covariate and an average neuropathology score because cases and controls are not straightforward to determine in these population studies, as discussed in response #13 below).

In our study, we chose to leave our expression profiles uncorrected for all covariates, including PMI. Our reasoning is that we instead allow MD-AD to learn from the available gene expression patterns, and subsequently assess how these covariates interact within our final models. **In that way, we don't need to rely on correction methods which may fail to fully remove the effect of a confounder and leave unexpected residual effects.** Using our method, as shown in **Supplementary Figure 4**, some nodes do have residual correlations with PMI, but PMI certainly does not appear to be a driving force in our model.

Finally, we performed some additional analyses to ensure that these gene expression covariates were not an important factor in our prediction performance results presented via our cross-validation evaluations. To that end, we use the following method to correct our gene expression data for sequencing-related covariates: we linearly regressed PMI and RIN from our expression inputs by modeling the expression of each gene as a linear regression with PMI and RIN. We then saved the residuals of the predicted expression value as our corrected expression values.

We then performed the same model training and evaluation procedures as described in our paper using these corrected gene expression values, and found that these results were quite similar to our original results with uncorrected gene expression values. We have added this additional analysis to our Supplementary Methods (lines 193-213), and results are shown in (newly added) **Supplementary Figure 11a-b**. Together, these findings indicate that, although PMI does seem to have effects on gene expression values, we do not see a large impact of this potential confounder on our final results, nor do we see MD-AD heavily relying on variations in PMIs.

12. ACT, MSBB, and ROSMAP cohorts were used for training, but which cohorts were used for predictions discussed on p. 4-5? From the supplementary materials (p3) it seems that each dataset was split into 5 training and test sets, but that means the training and test sets are only semi-independent. Please explicitly state this (on p4 or 5). Also, it is important to mention that there was no sample overlap in these cohorts, this can be added in the main text. Only on p5 the ultimate out of sample prediction is mentioned, but prediction should always be out of sample to avoid bias, so please discuss why in-sample prediction (i.e. prediction in the sample that is also used for training) is chosen as well, as this is generally not considered good scientific practice

We thank the reviewer for raising these concerns. We think the reviewer misinterpreted our use of the word “in-sample” (see below), as our training and test sets are indeed independent from each other. As illustrated in **Supplementary Figure 1e**, we split the data into five independent groups (where each group acted as a test set once). For each instance, we kept the test set separate from the other samples, and used the remaining 4/5ths of the samples as a training set, on which we performed 5-fold cross validation. We then selected a set of hyperparameters based on CV performance, re-train an MD-AD model with these hyperparameters on the full training set (4/5 of samples), and then finally report performance on the test set (1/5 of samples) which had been left out during the training and CV process. As the reviewer suggested, we clarified that there is no overlap between cohorts in the main text (line 118; see excerpts below).

In order to provide robust performance metrics, we did not use a single train-test split, and instead performed the exact same procedure described five separate times, with each 1/5 of the data acting as a test set once. In that way, we are able to obtain multiple “test performance” metrics after performing CV and training on each test group’s respective training set. Thus, we would like to emphasize that our

reported prediction performance is based on samples that were *not* used to train an MD-AD model. This may be a slight misunderstanding about our meaning for “in-sample” versus “out-of-sample” predictions: by “in-sample”, we meant predictions for test samples within the same cohort (*not* the same samples used for training), and by “out-of-sample” we meant predictions for samples in entirely separate cohorts. We thank the reviewer for raising these issues, and have clarified this in our paper in the following ways, which are shown below for your convenience:

- We edited **Supplementary Figure 1e** (previously 1d) to more clearly illustrate our model selection and evaluation process:

Supplementary Figure 1e

- We added a detailed overview of our five iterations of training (with CV) and testing in the Supplementary Methods. In particular, we revised and expanded this subsection to clarify that test samples used for evaluating a particular model are never also used for training it (Supplementary Methods lines 151-163):

Cross-validation and model tuning. For our model training and evaluation, we use a modified cross-validation and testing scheme as illustrated in Supplementary Figure 1e, in which we perform five separate rounds of model tuning with cross-validation (CV) followed by evaluation in a test set. For a single round, one-fifth of all samples are assigned to a held-out test set. Then using the remaining 4/5ths of the samples, we perform 5-fold cross-validation to select hyperparameters with the best prediction performance. We then train the selected model using the full training set (4/5ths of the original data) and then report performance on the held-out test set. In order to evaluate the robustness of our evaluation metrics under different splits, we initially split the full dataset into 5 separate groups, and repeated the above process 5 total times, where each one-fifth of the data acted as a held-out test set once. We note that across these iterations, different training sets selected different configurations of hyperparameters, and for each train/test round we trained the full training set on the specific

configuration selected by cross-validation in that training set. Thus, our test set evaluations (e.g., in Figure 2a) reflect average test performance for the selected models in each round.

- We clarified that predictions discussed on p. 4-5 were based on training/test splits and refer the reader to the Supplementary Methods section for details (lines 140-143):

*In the first pass at model evaluation, we **trained** MD-AD using standard five-fold cross-validation (CV), **and assessed** the average $1-R^2_{CV}$ error (mean squared error divided by the phenotype's variance in the test set) on the **held-out** test samples (Figure 2a). **Our hyperparameter tuning and evaluation procedures are described in detail in the Supplementary Methods and Supplementary Figure 1d.***

- Clarified that there was no sample overlap between the three cohorts (lines 116-118):

*In particular, the model is trained on expression data from the ROSMAP^{6,19,20}, ACT²¹ and MSBB²² cohort studies, which together have 1,758 gene expression profiles for 925 distinct individuals (**with no participant overlap between cohorts**).*

13. p4 line 116 and p6 lines 168-169 the sample sizes are mentioned, but this is not split into cases and controls. Please provide numbers for cases and controls separately

We thank the reviewer for the suggestion. However, our data comes from population studies, not case/control cohorts, so due to the complex nature of AD-related neuropathology, defining cases vs. controls is not always straightforward. For example, some individuals have high rates of plaques but not tangles, or vice versa. Thus, because some studies provide detailed neuropathological measures across several variables rather than binary labels, we are unable to list these numbers for our three main cohorts. Instead, we've added **Supplementary Table 2** which provides summary information about each cohort. For the external validation studies (originally) mentioned in lines 176-179, binary labels were provided, so we have added them as suggested, shown below for your convenience:

*Next, as the ultimate test of MD-AD out-of-sample predictions, we assessed performance on three independent studies never seen by the model: Mount Sinai Brain Bank Microarray (MSBB-M; $N=1,047$; **565 AD cases and 482 controls**), Harvard Brain Tissue Resource Center (HBTRC; $N=338$; **246 AD cases and 92 controls**)⁷, and Mayo Clinic Brain Bank ($N=157$; **81 AD cases and 76 controls**)²⁶.*

14. p5 lines 163-164 states 'we observed that adding new samples improved performance for a phenotype even when the phenotype in question was not measured in the new samples' The authors explain this by saying that the shared representation learned by the algorithm captures the underlying biological signal - but how? And is this not dependent on the correlation between the phenotypes (and thus not a general effect)?

We thank the reviewer for the question. What we meant is that, by adding additional samples (even if they do not have all phenotype labels), the representation learned by the hidden layers of the network is improved, which affects the model's predictive ability for phenotypes (e.g. **Figure 2b**). This improvement *does* rely on our assumption that there exists some shared underlying biological phenomena relevant to all phenotypes. Promisingly, we observe that MD-AD is able to improve its overall hidden representation with the addition of new cohorts (despite noise and differences in covariates and phenotype measurements, e.g., in **Supplementary Table 2**), which is evidenced by the improvement across *all* phenotypes even when the added samples only provide *some* phenotype labels. We have added brief clarification to this sentence to highlight that the MD-AD's representation is improved by its access to more samples (lines 172-175):

*This suggests that the shared representation learned by MD-AD (**which is improved by access to additional sparsely labeled samples**) captures the underlying biological signal common across noisy neuropathological phenotype measurements.*

15. An implicit assumption seems to be that gene-expression patterns are causally related to AD pathology. However, the reverse may also be true: that due to the forming of plaques or the use of medication, DNA methylation patterns are altered, and differences in gene expression are a consequence rather than a cause of the disease, which is difficult to resolve using post-mortem transcriptomic data, although the mouse data can shed some light on this. This can be briefly discussed.

We thank the reviewer for raising this point. We completely agree that implying causality from our results would be problematic because we cannot infer causal relationships purely based on observational data. Throughout our manuscript, we have carefully considered our wording to ensure that causality is not implied by our results. However, we do argue that identifying (causal) drivers of neuropathology is an important goal, and our analyses were intended to contribute to this goal. By investigating the relationship between gene expression and neuropathology, we are able to refine our understanding of *potential* drivers. By identifying meaningful interrelationships among genes and neuropathology, we provide a refined set of hypotheses for further investigation (e.g., in mechanistic studies).

As suggested by the reviewer, we have made explicit that we do *not* assume any casual relationships from our analyses and added this limitation to the discussion section (lines 389-392). An excerpt of this revision is provided below for your convenience:

We note that like any other machine learning-based model applied to observational data, we are unable to directly infer causality in our framework, as both gene expression and neuropathology data used for MD-AD were collected from post-mortem brains. Nevertheless, with our list of results and our careful evaluation of sex effects we now have an important new road map with which to guide our exploration of the role of microglia in AD in a sex-informed manner. This perspective will be critical not only for mechanistic studies whose results could be obscured by sex effects but also, more importantly, by guiding the study design of clinical trials as highly targeted therapeutic agents emerge to modulate the immune system in AD.

16. p8 line 246: Briefly add what the Integrated Gradient algorithm entails and why it is appropriate

We thank the reviewer for the suggestion. Model interpretability methods have allowed researchers to gain insights about complex models' predictions with respect to input samples. Integrated Gradients (IG) is one of the most commonly used methods for interpreting deep neural networks which, given a single sample, estimates the importance of each input feature on the output(s). This approach has theoretical underpinnings and satisfies important axioms of machine learning attribution methods (Sundararajan et al., 2017). As described by the authors of the original paper, the IG score for a specific sample is calculated for each input dimension with respect to each output dimension by accumulating gradients along the path from the input to output. Thus, by using this algorithm for each gene expression sample, we are able to evaluate the importance of each gene on each phenotype prediction by the MD-AD model.

We agree that it is important to properly contextualize the Integrated Gradients algorithm before describing related results. We have added the following to introduce the reader to the Integrated Gradients algorithm:

- A new section in the supplementary methods section which describes the IG method (and how we apply it to MD-AD) more thoroughly, as summarized above (Supplementary Methods lines 309-321).
- A sentence to introduce the integrated gradients algorithm at lines 251-254, including a cross-reference to the supplementary methods section. For your convenience, we show the revised main text below:

*We next sought to interpret MD-AD's learned parameters to identify the set of genes (and their relationships) that underlie its impressive predictive performance. **Integrated Gradients (IG)¹⁸**, one of the most widely used interpretability methods developed for deep neural networks, estimates the importance of input features on a model's predicted output for a particular data point (See*

Supplementary Methods for details). Here, we applied the IG algorithm on the fully trained model in an ensemble fashion to ensure robustness (Supplementary Methods, Supplementary Figure 6), producing an “importance score” for each gene (Supplementary Table 4).

Sundararajan, M., Taly, A. & Yan, Q. Axiomatic attribution for deep networks. *34th Int. Conf. Mach. Learn. ICML 2017* 7, 5109–5118 (2017).

17. lines 260-263: Gene importance scores tended to yield more specific pathway enrichment than scores based on correlations - can the authors state whether they think ‘more specific’ means ‘closer to the truth’ (as it seems to suggest now) and if so why? This statement needs an interpretation by the authors here

We thank the reviewer for the question. While future advancements in AD research will be needed to confirm whether MD-AD’s results are indeed “closer to the truth”, we think the shorter list of 46 enriched pathways from MD-AD may reveal more fundamental—and thus, potentially more informative—specific processes. In contrast, the correlation-based rankings highlight 180 pathways which span many smaller processes and thus may be less informative for future researchers.

We have edited the sentence to highlight that a more specific set of pathways may be more informative for future research (lines 269-271):

Overall, gene importance scores generated via correlations alone were enriched for more REACTOME pathways (Supplementary Figure 7b), whereas MD-AD offered a more specific set of processes for further investigation (Figure 5b).

18. lines 260-263: How was the ‘gene correlational score’ calculated? Please explain

We thank the reviewer for the question. As the reviewer recognized, computing a correlation-based ranking for the genes is complicated by the use of six phenotypes. In order to generate correlation-based rankings (for comparison with the MD-AD gene rankings), we computed correlation coefficients between each gene’s expression level and each of the six training phenotypes. We then ranked the genes based on their correlation coefficients and then averaged the rankings across the six phenotypes to obtain a final correlation-based gene ranking.

We have expanded our explanation of our correlation-based rankings in our Supplementary Methods (lines 396-402) and added a reference to it in the main text at our first mention of correlation-based rankings (line 263). The expanded description of correlation-based rankings (Supplementary Methods lines 396-402) is provided below for your convenience:

For comparison with a linear gene ranking method, we also generate correlation-based gene rankings as follows: we calculate the correlation coefficients between each gene's expression level and each neuropathological phenotype (across all samples in our dataset), and then percentile-rank the genes by their average correlation coefficients across all six phenotypes (with 0 for the most negatively correlated and 1 for the most positively correlated gene with high pathology). Our final correlation-based gene ranking is the average over the phenotype-specific rankings.

19. lines 272-273: 'Of the ... correction' Here it is important to know whether this analysis was based on IG scores that were calculated in the total sample, or separately in males and females, thus capturing a fixed effect of sex or looking at less systematic differences (the latter). Also, were expression profiles of genes on the X-chromosome included?

We thank the reviewer for the questions. In the following response (and reflected in our manuscript), we clarify our methods for evaluating differential importance of genes between sexes in the MD-AD model by first mentioning that our method is based on an interaction model (in the main text), and second, by elaborating on this method (with corresponding additions to the supplementary methods). Third, we provide missing details about the chromosomal locations of the genes used in MD-AD.

First, regarding original lines 272-273 (281-282 in revised manuscript), we would like to make a clarification, as we may have caused some confusion about how 'differential MD-AD importance between sexes' was evaluated. Our IG analysis provides *per-sample* gene importances for predicted neuropathology variables. When we compute whether there is a significantly different gene importance between males and females, we use an interaction model (described in lines 279-281) and evaluate whether there is a significant coefficient for the interaction term (i.e., whether sex affects the slope of the relationship between the gene's expression and its importance). We note that all IG scores are calculated from an MD-AD model trained on the full dataset (both males and females), but our interaction model evaluates whether the MD-AD model differently associates a gene's expression with neuropathology based the sample donor's sex. We hope this clears up any concern for the reviewer, and have added some clarification to the sentence that we use per-sample IG scores, and identify *differential* importance between sexes using an interaction model. For your convenience, the updated text is shown below (lines 279-283).

In particular, to identify sex-interacting genes relevant to AD we modeled each gene's per-sample IG score as a linear combination of the gene's expression, the individual's sex, and the interaction between them. Of the 14,591 genes in our dataset, 6,465 showed differential MD-AD importance between sexes in an interaction model ($p < 0.05$ after FDR correction), demonstrating that sex-specific expression effects in AD may be widespread.

We have additionally added some clarification to the Supplementary Methods to describe how per-sample IG scores are calculated, and how they are used to compute interaction effects. The expanded text is shown below for your convenience (Supplementary Methods lines 409-415):

Generating sample-level gene importances scores. *To simplify our analyses, we generate consensus IG scores for each gene within each sample as follows: for each sample and gene, we average over the gene's IG weights across both neuropathological phenotypes and runs in order to obtain its average importance for general neuropathology across all runs.*

Measuring interaction effects. *To monitor the presence of interaction effects in gene importance scores, we modeled the consensus per-sample IG scores as a linear combination of a gene's expression level, a covariate of interest, and the interaction of the two. ...[rest of subsection unchanged from original submission]...*

Finally, we thank the reviewer asking about sex chromosome versus autosomal genes in our dataset, as we realize we neglected to mention this information in our original manuscript. For our analyses, we consider all genes which are available across ACT, ROSMAP, and MSBB datasets, and keep all genes with at least 1/3 non-null observations, with no other exclusionary criteria. Our resulting dataset contains 14,591 genes, of which 96.3% are autosomal (3.5% on only the X chromosome, 0.1% on only the Y chromosome, and 0.1% on both X and Y chromosomes). Although we consider all genes for our analyses of differential importance between sexes, we found that there was not a significant difference between sex and autosomal chromosomes in their rates of significance for differential importance between sexes ($\chi^2 = 1.2$, $p=0.75$), so the presence of many genes with differential importance between sexes does not appear to be driven by the sex chromosome genes. Additionally, we were wondering if there was an over-representation of sex chromosome genes in our top MD-AD gene list (i.e., **Figure 4**). In general, we see no significant difference in gene scores between sex and autosomal genes ($t=0.015$, $p=0.988$). We hope this investigation into our sex chromosome vs. autosomal genes lessens any potential concerns about the inclusion of sex chromosomes in our analyses. We have added details about the chromosomal locations of genes to our Supplementary Methods (lines 22-25), and provide the added sentence for your convenience below.

We then combined the gene expression datasets and kept all 14,591 genes that are present across all three datasets. Of these genes, 96.3% are autosomal (3.5% on only the X chromosome, 0.1% on only the Y chromosome, and 0.1% on both X and Y chromosomes).

20. Table S4: I am bit surprised to see that the APOE gene has such a low MD-AD consensus score, while it is one of the strongest genetic effects on AD - how would the authors explain this?

We thank the reviewer for raising this point. First, we would like to note that, while our consensus scores range from 0 to 1, with high scores representing a strong positive relationship between a gene and

neuropathology, ‘low’ scores close to 0 are also indicative of important genes, as they indicate a strong negative relationship between genes and neuropathology. Thus, while APOE had a low consensus score, it’s actually at the 91st percentile of genes when ranked by negative neuropathology-related importance. We have updated the Supplementary Methods (lines 383-391) to clarify that both low and high scores reveal important genes in the MD-AD framework, which we show below:

We next aggregate our IG scores into gene rankings by calculating the ranks of each gene (for each phenotype) in each run, and then averaging across runs to obtain consensus gene ranks. For each phenotype (see Supplementary Figure 6 for illustration). Thus, the gene with the highest consensus IG score (i.e., score close to 1) is the gene with the highest average rank across runs (most positively associated with the neuropathological phenotype), and the gene with the lowest consensus IG score (i.e., score close to 0) is the gene with the lowest average rank across runs (most negatively associated with neuropathology).

Interestingly, while APOE’s MD-AD ranking was relatively high, APOE’s correlation-based score was 0.447 (quite low in terms of absolute rank), and it generally had a weak correlation with neuropathology phenotypes. However, we don’t find this to be too surprising. Although APOE *genotypes* have been strongly tied to AD onset and severity, gene expression studies have not agreed on a definitive relationship between APOE’s *expression* and AD. Instead, it appears that the relationship between APOE gene alleles and AD may be mediated by other factors, such as expression levels of other regulatory genes (Rhinn et al., 2013) or epigenetic effects driven by DNA methylation (Lee et al., 2020), thus weakening the relationship between APOE’s expression level and AD neuropathology.

Rhinn, H., Fujita, R., Qiang, L. et al. Integrative genomics identifies APOE ϵ 4 effectors in Alzheimer's disease. *Nature* 500, 45–50 (2013). <https://doi.org/10.1038/nature12415>

Lee, Eun-Gyung, et al. "Redefining transcriptional regulation of the APOE gene and its association with Alzheimer’s disease." *PLoS one* 15.1 (2020): e0227667.

21. Using multiple data-sets was there some sort of data processing built into a pipeline to ensure data integrity? (Was there a re-scaling of the data done? What happened to outliers?)

We thank the reviewer for the question. Because the datasets we used were provided through the AMP-AD consortium, extensive pre-processing and quality control filters were applied by the researchers who collected and published initial findings from the data, and thus we did not perform additional filtering (e.g., removing outliers). However, we did apply some additional re-scaling and normalizing so that the datasets would have similar scales before combining them with batch correction. We have added these details to the Supplementary Methods (lines 12-26), along with a reference to find all pre-processing and quality control information from the original papers, which we show below for your convenience.

In all three studies, extensive quality control measures were taken during the original processing of the datasets, as described by the original papers introducing transcriptomic datasets for ACT¹, MSBB², and ROSMAP⁵ cohorts. All samples which passed quality control checks in these individual studies were included in our study. If Ensemble gene IDs were provided, we mapped them to gene symbols to keep consistent gene identifiers across datasets. In order to compile gene expression samples across the three cohorts, we retain expression levels for genes which are present in all datasets. Within each dataset, we exclude genes with null values for over two-thirds of samples. For ACT and ROSMAP, gene expression measurements were provided in normalized FPKM units, so we log-transformed the RNA-Seq datasets to obtain gene expression datasets that were roughly normally distributed (whereas MSBB gene expression data were already normalized). Then, for all datasets, we normalized values such that each gene's expression measures varied between 0 and 1. We then combined the gene expression datasets and kept all 14,591 genes that are present across all three datasets. Of these genes, 96.3% are autosomal (3.5% on only the X chromosome, 0.1% on only the Y chromosome, and 0.1% on both X and Y chromosomes). Finally, we performed batch effect correction with ComBat to reduce systematic differences across studies (Supplementary Figure 1c)⁶.

22. Typos:

- p3 suppl. materials: hyperparamters should be hyperparameters
- line 91 (figure) 1d should be (figure 1d)

We thank the reviewer for the catch. These typos have been corrected.

Reviewers' Comments:

Reviewer #1:

Remarks to the Author:

The authors have carefully addressed most of my comments.

1. I have still a concern regarding the pooling of data from several studies, which might be clinically rather different. The authors have added a Table, which addresses this aspect up to a certain extend, but I think that rather important information is still missing, such as: APOE4 status (one of the main risk factors for AD) and cognition scores, e.g. MMSE.

The authors argue that their empirical results do not support the concern that a model trained with pooled data is biased towards the most abundant cohort. Do the authors have any explanation for such a rather counter-intuitive finding?

In that line: How would results change, if MD-AD was trained on ROSMAP + only a fraction of MSBB (and then tested on ATC)? At some point the cohort bias should become visible.

2. page 6, line 183 and following: I think a similar analysis should be done for APOE4 status.

Reviewer #2:

Remarks to the Author:

The authors are commended for such an elaborate revision, they have addressed all comments satisfactorily. I have two remaining small points:

Comment 2 - I agree with the response, yet it is unclear what has been changed in the manuscript - could you please add 1 sentence to the manuscript reflecting your response?

Comment 10 - thanks for the clarification - it would be nice to also be explicitly mention this in the manuscript (can be brief)

well done!

Thank you for considering our manuscript “Unified AI framework to uncover deep interrelationships between gene expression and Alzheimer’s disease neuropathologies” for review by *Nature Communications*, and for allowing us to submit a point-by-point response and incorporate the comments that we have received into a new revision of the manuscript.

We would also like to thank the reviewers for their careful consideration of this manuscript and suggestions for improvement. Below is a point-by-point response to the referee comments. The reviewer’s comments are in blue and italics, and our replies are in black. We have additionally included some excerpts from our revised manuscript for your convenience.

Reviewer #1 (Remarks to the Author):

The authors have carefully addressed most of my comments.

We thank the reviewer for their detailed comments and suggestions that substantially improved our paper in the previous round. We have addressed their remaining concerns below.

1a) I have still a concern regarding the pooling of data from several studies, which might be clinically rather different. The authors have added a Table, which addresses this aspect up to a certain extend, but I think that rather important information is still missing, such as: APOE4 status (one of the main risk factors for AD) and cognition scores, e.g. MMSE.

We thank the reviewer for the suggestion and raising these valid concerns. In order to more completely address the concerns, we have split question 1 into parts a-c (see below). First, we describe updates to Table 2 which shows inter-cohort differences. Despite these substantial differences between groups, we highlight in parts b and c below that the benefits of increased sample size (even from different cohorts) outweigh potential drawbacks of bias introduced by additional samples, supported by additional experiments we performed in response to this comment.

First, we have updated Table 2 as much as possible as suggested by the reviewer, as described below:

- We note that APOE status was provided in Table 2 in our previous revision (6th row), but recognize that our labeling for it may have been unclear. We have updated the row title to highlight that the numbers are referring to alleles (i.e., from “22/23/24/33/34/44” to “ $\epsilon 2\epsilon 2/\epsilon 2\epsilon 3/\epsilon 2\epsilon 4/\epsilon 3\epsilon 3/\epsilon 3\epsilon 4/\epsilon 4\epsilon 4$ ”). Although APOE $\epsilon 4$ status is contained in that row (including counts of heterozygous vs. homozygous carriers of $\epsilon 4$), we have added an additional row specifically comparing the what percentage of samples are from people with at least one $\epsilon 4$ allele (see newly added row 7).
- Regarding cognition scores, each study had a different neuropsychological battery, and we only had access to raw test scores from ROSMAP. Consequently, we are unable to directly compare cognition across all cohorts beyond a final diagnosis of dementia (i.e., row 8 in Table 2). However, we have identified some slightly more granular categories of cognition status, which

we have added to Table 2. In particular, we have added counts of diagnoses based on NINCDS/ADRDA criteria (available for ACT and ROSMAP; new row 9) and counts of cognitive status which include mild cognitive impairment (available for MSBB and ROSMAP; new row 10). However, we note that MSBB and ROSMAP each had their own method for quantifying mild cognitive impairment (Clinical Dementia Rating in MSBB; consensus reached by a neuropsychologist and clinician based on several cognitive test scores in ROSMAP).

TABLE 2 (Revised)		ACT	MSBB	ROSMAP
Row 6	APOE genotype (% ε2ε2/ε2ε3/ε2ε4/ε3ε3/ε3ε4/ε4ε4)	0.0/8.0/1.3/70.5/18.9/ 1.3 ^{MMM R}	1.2/11.7/0.7/55.0/29.2/ 2.1 ^{AAA RR}	0.9/13.1/2.2/61.1/21.8/ 0.9 ^{A MM}
Row 7	APOE ε4 carrier (% carrying at least 1 ε4 allele)	21.47 ^{MM}	32.03 ^{AA R}	24.91 ^M
Row 8	Cognition status (% dementia)	48.19 ^{MMM}	59.27 ^{AAA RRR}	42.80 ^{MMM}
Row 9	Cognition status including mild impairment* (% None/Mild Impairment/Dementia)	NA	29.5/11.3/59.3 ^{RRR}	31.7/25.5/42.8 ^{MMM}
Row 10	Diagnosis based on NINCDS/ADRDA criteria** (% No dementia/Probable AD/Possible AD/Dementia-type unknown)	51.3/18.7/20.5/9.5 ^{RRR}	NA	57.2/35.6/5.0/2.2 ^{AAA}

*For MSBB, this was based on the clinical dementia rating (CDR). For ROSMAP, this was based on consensus between a neuropsychologist and clinician based on a battery of tests. **For NINCDS/ADRDA criteria, Possible and Probable AD both indicates that the individual has dementia, but with differing certainty about the cause while they were alive.

1b) The authors argue that their empirical results do not support the concern that a model trained with pooled data is biased towards the most abundant cohort. Do the authors have any explanation for such a rather counter-intuitive finding?

As a point of clarification, we do *not* argue that there does not exist *any* bias towards the more abundant cohorts. Instead, we argue that the benefit of increasing available samples (even from a different cohort) outweighs the drawbacks of having biases introduced by cohorts with different distributions.

Our study assumes that there exists a meaningful relationship between gene expression and AD neuropathology, and despite some demographic differences between cohorts or gene expression measurement effects, there should still remain some underlying signal that is consistent across all cohorts which can be learned by our MD-AD model. This was originally supported by our findings shown in **Figure 2b**, which was carried out specifically for ROSMAP samples because that is the only study for which we can observe changes in performance across *all six* phenotypes. However, to more thoroughly demonstrate how adding cohorts affects prediction performance, we have now added the same experiments for ACT and MSBB test samples as **Supplementary Figure 3a (shown below)**. However, we are only able to observe prediction performance for neuropathological phenotypes which exist in both the training and test set for a specific experiment, and thus they may not provide a full picture of how predictive ability of the model changes based on available training samples.

Supplementary Figure 3a supports our claim that pooling data from cohorts lead to improvements in performance *despite* some covariate shifts. First, we note that more abundant datasets *do* tend to have

lower prediction error when comparing within a single phenotype for the pooled data model (shown in **Supplementary Figure 3a** on the furthest right diamond marker in each scatter plot). This may imply that there *are* some biases towards the more abundant data sets. Furthermore, the most abundant dataset (MSBB) does not improve when new datasets are added to it (however, it's important to note that we only see performance for three of six phenotypes, and these tend to be “easier” phenotypes to predict, according to their predictability in other cohorts). Nevertheless, for all neuropathological phenotypes in all cohorts, adding samples from the two other cohorts leads to prediction performance that is better than or about equal to predictive performance of a model trained on only samples from a that single cohort alone (shown in diamond-shaped markers in **Supplementary Figure 3a**). Together, these results imply that, despite potential biases introduced by adding samples from a separate cohort, increased sample sizes tend to lead to improved (or at least similar) prediction performance for all cohorts.

1c) In that line: How would results change, if MD-AD was trained on ROSMAP + only a fraction of MSBB (and then tested on ATC)? At some point the cohort bias should become visible.

We thank the reviewer for suggesting this analysis to further clarify how the addition of samples impacts prediction performance, particularly when transferring across cohorts.

First, As Reviewer 1 suggests, and consistent with the fact that the cohorts have some substantial differences (Table 2), models trained with samples from separate cohorts (i.e., transferred models) tend to do worse than models which are trained at least partially using samples from the same cohort (i.e., augmented models). This is highlighted by the tendency for higher error for circular versus diamond markers in **Supplementary Figure 3a**, which represent transferred models versus augmented models (where training from a single dataset is augmented with samples from other datasets), respectively.

As suggested by the reviewer, we evaluated the transfer performance of MD-AD trained on ACT or ROSMAP with fractions of MSBB (the most abundant dataset) and tested the remaining dataset. When training on ROSMAP samples with increasing fractions of MSBB samples introduced, we see relatively stable prediction performance for ACT test samples (either very slightly increasing or slightly decreasing as more MSBB samples are added) (**Supplementary Figure 3b**, right). This may indicate that both ROSMAP and MSBB have distributions that are somewhat different from ACT and thus may not transfer very effectively. Nevertheless, we do see that when *adding* these cohorts to a model that is also trained with ACT training samples, the additional samples do tend to slightly improve test prediction performance (perhaps by adding additional stability to training) (**Supplementary Figure 3a**, middle). When we consider the same experiment for ROSMAP (this time training with ACT samples and increasing fractions of MSBB samples), we find that for many phenotypes, the addition of MSBB samples improves transfer performance for ROSMAP. This may indicate that MSBB and ROSMAP have a more similar relationship between expression and neuropathology than they do with ACT. However, we think it is also likely that the improved performance in this case may also be due to the fact that ACT is the smallest dataset, so adding MSBB samples has a dramatic effect of increasing sample sizes available for training.

Together, these results indicate that there likely *are* biases introduced when training with samples from different cohorts (as demonstrated by worse performance for transferred vs. augmented models as

Supplementary Figure 3. (a) Test prediction performance using 5-fold cross-validation when training on different subsets of available datasets. We display performance for each dataset’s test samples separately. Circle markers show performance when transferring a trained model to a new dataset; diamond markers show changes in performance when augmenting the training set with samples from other datasets. **(b)** Same analysis as part (a), but highlighting how transfer performance changes when adding additional MSBB samples during training.

described above). Nevertheless, we've demonstrated that across cohorts, pooling the datasets together seems to have an overall positive effect on prediction performance despite some potential biases.

Along with adding **new Supplementary Figure 3 above**, we have added the following discussion of these experiments to our manuscript:

Revised manuscript line 176:

Because our model was trained and evaluated on ACT, MSBB, and ROSMAP datasets, we assessed whether residual (uncorrected) batch effects affected performance. [...] Third, we observed that adding new samples improved performance for a neuropathological phenotype even when the phenotype in question was not measured in the new samples (see gray footprints around markers in Figure 2b). ***The same analysis repeated with the other two cohorts as test sets revealed similar findings (Supplementary Methods, Supplementary Figure 3).*** This suggests that the shared representation learned by MD-AD (which is improved by access to additional sparsely labeled samples) captures the underlying biological signal common across noisy neuropathological phenotype measurements.

Revised Supplementary Methods lines 186-197:

Next, in order to evaluate the contributions of each dataset to prediction performance, we performed the above procedure with different subsets of available datasets. Because ROSMAP is the only dataset with all available neuropathological phenotypes, we evaluate performance specifically on ROSMAP. In Figure 2b, we show ROSMAP test samples' MSE performance when trained on all subsets of ACT, MSBB, and ROSMAP training samples (following the same cross-validation procedure described above). ***We additionally repeated the same analysis using MSBB and ACT test samples and computed their prediction performance for available phenotypes (Supplementary Figure 3a). In order to evaluate how transfer performance (i.e., training and evaluating with samples from disjoint cohorts) was impacted by the addition of samples, we performed an additional analysis where we trained with ACT samples and varying fractions of MSBB samples to see how additional MSBB samples impacted ROSMAP test performance (and also evaluated the reverse, training on ROSMAP and MSBB and testing with ACT samples) (Supplementary Figure 3b). Interestingly, we saw that ROSMAP test performance improves with the addition of MSBB samples during training with ACT, whereas ACT samples generally do not improve with the addition of MSBB samples during training with ROSMAP. This may imply that there are more pronounced distributional differences for the ACT cohort when compared with other cohorts, or that improvements are more apparent when the training set is much smaller (as is the case when training with only ACT samples).***

2. page 6, line 183 and following: I think a similar analysis should be done for APOE4 status.

We thank the reviewer for the suggestion, which we interpreted as a suggestion to compare models' neuropathology predictions for APOE ϵ 4 carriers vs. non-carriers (rather than for those with diagnosed AD vs. controls). We have carried out the suggested analyses with available samples and find that results are similar to our findings with AD vs. control individuals, as displayed in new **Supplementary Figure 5**.

Compared with those without any $\epsilon 4$ alleles, those with at least one copy $\epsilon 4$ allele received higher neuropathology predictions across all models; however MD-AD's predictions showed the highest difference between groups ($t=5.9$, $p<.001$) (**Supplementary Figure 5a-b(left)**). This is similar to our findings with AD vs controls, however with a smaller but still significant effect size.

Similarly to our analyses comparing individuals with AD vs controls, the difference in predictions is most pronounced for younger individuals (under 75 years old) (**Supplementary Figure 5b(right)-c**). However, this may be in part due to the fact that many younger carriers of APOE $\epsilon 4$ alleles have higher rates of AD than non-carriers, and thus would be expected to have higher neuropathology predictions. When comparing among only those with AD, we see similar neuropathology predictions between APOE $\epsilon 4$ carriers and non-carriers for those that have AD (**New Supplementary Figure 5d-e**).

We have added a brief mention to this analysis on line 194 of the revised manuscript, and described it in more detail in the Supplementary Methods lines 271-277.

Revised Manuscript line 194:

As shown in Figure 2c, we observed a highly significant difference in predicted neuropathology scores between AD cases and controls (two-sided t-test: $t= 22.98$, $p<0.001$), and these differences were more pronounced for MD-AD compared to the other baseline models (results split by dataset are shown in Supplementary Figure 3a). More convincingly, when split by age group (Figure 2c right panel), we consistently observed a significant increase in predicted neuropathology for AD vs control samples, but the difference was largest in individuals under 75 (between-groups p-values are shown in Supplementary Figure 3b. ***The same analysis comparing APOE $\epsilon 4$ carriers to non-carriers revealed a similar pattern, shown in Supplementary Figure 5.***) This is consistent with the observation that aging individuals who are cognitively non-impaired often have substantial neuropathology²¹. Together, these results indicate that MD-AD can identify generalizable gene expression patterns that are predictive of AD-related neuropathology across varied age ranges, and thus it is unlikely that these patterns merely capture normal aging.

Revised Supplementary Methods lines 271-277:

Figure 2c shows that MD-AD provides the largest differences in neuropathology scores between individuals with and without neuropathological diagnoses of AD. We further compared neuropathology scores between AD and non-AD individuals split by age group (significance between groups shown in Supplementary Figure 3b). ***A similar analysis was carried out comparing carriers of the APOE $\epsilon 4$ allele with non-carriers (instead of AD vs control individuals). Results shown in Supplementary Figure 5a-c revealed similar patterns, including improved discrimination between groups for MD-AD compared with MLP and linear baselines, and more pronounced differences in predicted neuropathology for younger individuals APOE $\epsilon 4$ carriers versus non-carriers. However, when comparing predicted neuropathology between APOE $\epsilon 4$ carriers versus non-carriers within the same cognitive diagnosis, we do not see a difference in predicted neuropathology for AD-afflicted APOE $\epsilon 4$ carriers and non-carriers (Supplementary Figure 5d-e).***

Supplementary Figure 5. External validation results considering APOE status. (a-c) show how neuropathology predictions compare for carriers vs. non-carriers of the APOE $\epsilon 4$ allele. (a) t-test statistics measuring differences between each model's predicted neuropathology scores for carriers vs. non-carriers of APOE $\epsilon 4$. (b) For samples from external validation data sets, we obtain neuropathology scores for each sample from each model. *Left*: Box plots displaying the distribution of predicted neuropathology scores from each method for APOE $\epsilon 4$ carriers vs. non-carriers. T-tests highlight between-group differences for each method (2-sided t-test, *** $p < .001$) (Elements: center line, median; box limits, upper and lower quartiles; whiskers, 1.5x interquartile range from quartiles; points, outliers). *Right*: Box plots displaying the distribution of MD-AD's predicted neuropathology scores split by age group and APOE status. (c) sample sizes and significance levels of t-tests comparing pairs of groups shown in part b. (d) *Left*: Predicted neuropathology scores for each method split by diagnosis (replicated from Figure 2c). *Right*: Box plots displaying the distribution of MD-AD's predicted neuropathology scores split by both diagnosis and APOE status. (e) sample sizes and significance levels of t-tests comparing pairs of groups shown in part d.

Reviewer #2 (Remarks to the Author):

The authors are commended for such an elaborate revision, they have addressed all comments satisfactorily. I have two remaining small points:

We thank the reviewer for their detailed suggestions in the last round, and for the positive feedback! We have addressed their remaining small points below.

Comment 2 - I agree with the response, yet it is unclear what has been changed in the manuscript - could you please add 1 sentence to the manuscript reflecting your response?

We thank the reviewer for the suggestion. We have updated the following section in our manuscript to clarify that MD-AD could accommodate datasets with non-overlapping phenotypes:

Page 5, line 137:

During training, the MD-AD model continually updates model parameters via backpropagation, but only for labeled phenotypes from a given sample. Thus, for each phenotype for a given sample, MD-AD updates parameters from associated separate layers along with all shared layers. This lets us train a unified model from all available samples despite having many missing labels. ***Although in our application neuropathological phenotypes overlapped across datasets, MD-AD could accommodate non-overlapping phenotypes from different cohorts (as long as they are believed to be closely related and share a common underlying gene expression basis).*** Details of the MD-AD framework, modeling assumptions, and hyperparameter tuning are provided in Supplementary Methods.

Comment 10 - thanks for the clarification - it would be nice to also be explicitly mention this in the manuscript (can be brief)

We thank the reviewer for the suggestion. We have updated the manuscript to explicitly mention how correlation patterns among inputs are handled by deep neural networks:

Page 3, line 85:

*Additionally, the increased sample size from combining cohorts (in our case, doubling the number of samples available from any individual study) facilitates using deep learning models, which are expressive and able to capture complex non-linear interactions among features. **By composing layers of functions, deep neural networks collapse correlation patterns present in input data at intermediate layers in a way that is useful for prediction**¹⁴. In particular, multi-layer perceptrons (MLPs) have been used to effectively perform disease classification and prediction from gene expression data¹⁵⁻¹⁷.*

Reviewers' Comments:

Reviewer #1:

Remarks to the Author:

The authors have addressed all my questions and concerns.